# Persistent Topological Features in Large Language Models

## Abstract

Understanding the decision-making processes of large language models (LLMs) is critical given their widespread applications. Towards this goal, describing the topological and geometrical properties of internal representations has recently provided valuable insights. For a more comprehensive characterization of these inherently complex spaces, we present a novel framework based on zigzag persistence, a method in topological data analysis (TDA) well-suited for describing data undergoing dynamic transformations across layers. Within this framework, we introduce persistence similarity, a new topological descriptor that quantifies the persistence and transformation of topological features such as $p$-cycles throughout the model layers. Unlike traditional similarity measures, our approach captures the entire evolutionary trajectory of these features, providing deeper insights into the internal workings of LLMs. As a practical application, we leverage persistence similarity to identify and prune layers, demonstrating comparable performance to state-of-the-art methods across several benchmark datasets. Additionally, our analysis reveals similar topological behaviors across various models and hyperparameter settings, suggesting a universal structure in LLM internal representations.

## 1 Introduction

Large Language Models (LLMs) have revolutionized natural language processing by achieving unprecedented performance levels across a wide range of tasks (see Raiaan et al. (2024) for a review). Despite their success, the black-box nature of these models has raised significant concerns about interpretability and transparency (Liao & Vaughan, 2023). Moreover, their large scale demands a considerable amount of computational resources (Samsi et al., 2023; Bai et al., 2024), making it essential to reduce their size without compromising performance (Ma et al., 2023; Gromov et al., 2024; Men et al., 2024).

One strategy for addressing these issues has been to study the models' internal representations. Early works (Zeiler & Fergus, 2014) demonstrated that visualization techniques can effectively uncover hierarchical representations within convolutional neural networks, highlighting how lower layers focus on edge detection while higher layers correspond to object parts and semantic concepts. Additionally, (Olah et al., 2018) illustrated that analyzing weight matrices and neuron activations can reveal interpretable features and organizational structures within deep networks, providing insights into how complex patterns are encoded and processed.

More recently, geometric studies made progress by introducing concepts like intrinsic dimension to characterize the manifold of internal representations and its evolution across layers (Ansuini et al., 2019; Doimo et al., 2020; Pope et al., 2021). These methods have been successfully applied to transformer models in various works (Valeriani et al., 2023; Tulchinskii et al., 2024; Cheng et al., 2023). One notable achievement of this approach has been to show the emergence of semantic knowledge and abstraction phases in the middle layers of models, rather than at the final layers, as might be intuitively expected. However, these approaches provide only a static view of internal representations and suffer limitations in tracking their changes across layers.

A natural framework to address these limitations and to offer a more comprehensive characterization of the geometry of internal representations of neural networks is Topological Data Analysis (TDA). TDA is a set of unsupervised techniques that offers robust methods to describe the shape and structure of complex datasets. It has seen exponential growth with applications in computational biology

(Mandal et al., 2020), cosmology (Biagetti et al., 2021; Yip et al., 2024)], personalized medicine (Skaf & Laubenbacher, 2022), time-dependent data analysis (El-Yaagoubi et al., 2023), and machine learning (Hensel et al., 2021), just to name a few. One prominent tool within TDA is persistent homology, which tracks the birth and death of topological features across different scales, thereby capturing the multiscale behavior of a point cloud. Several studies have proposed persistent homology to investigate neural networks and their internal representations (e.g. Rieck, Bastian Alexander et al. (2023), Naitzat et al. (2020); Lacombe et al. (2021); Magai & Ayzenberg (2022)).

However, in the context of TDA applications, it has not yet been recognized that the internal representations of LLMs can essentially be viewed as dynamic point clouds evolving in time (layers). As pre-trained LLMs process inputs, they transform these point clouds within the representation space layer by layer, capturing essential features and relationships throughout the model's depth. Thus, it is natural to interpret these transformations as an evolving discrete dynamical system. To address this problem, we exploit a TDA tool developed to characterize time-varying point clouds and temporal networks, known as *zigzag persistence*.

Our approach achieves the following results:

- **ZigZag Framework for LLMs:** We build a framework to characterize the birth and death of topological features across transformer model's layers. As new contributions in the context of zigzag applications, we introduce the k-Nearest Neighbors-based filtration, and we interpret layers as time snapshots in a dynamic system, tracking the trajectory of features across layers.

- **Persistence Similarity:** We propose a new topological descriptor to measure which topological features persist across the layers of an LLM. Different than other similarity measures, persistence similarity tracks the entire trajectory of transformations between two layers.

- **Model Pruning:** As a showcase of our framework, we use persistence similarity as a criterion to prune layers without significantly degrading performance, finding comparable results to state-of-the-art methods.

- **Similarity of Results Across Models and Hyperparameters:** Our findings show similar results across different models, layers, and choices of hyperparameters of the framework. This suggests a degree of universality in the topological structure of LLM representations.

In summary, our framework presents a novel perspective by combining two fundamental elements: firstly, it provides a fine-grained geometric analysis of the internal representations through TDA; secondly, the zigzag persistence framework tracks the trajectory of topological features across layers. Distinct from traditional methods that solely compare representations at individual layers, our approach captures their entire evolutionary path, providing a richer understanding of how these features evolve and contribute to the model's decision-making processes.

## 2 RELATED WORK

**Geometry and Topology of Internal Representations.** The manifold hypothesis suggests that high-dimensional data often lies on a lower-dimensional manifold (Goodfellow et al., 2016). The estimation of this approximated manifold, known as intrinsic dimension, changes dynamically in deep networks, expanding and contracting in ways that impact performance (Ansuini et al., 2019), learnability (Pope et al., 2021), and the network's ability to generate flexible abstract data representations used for downstream tasks (Doimo et al., 2020), (Valeriani et al., 2023). Intrinsic dimension and neighbor composition analysis of internal representations of causal and masked transformer models helped in the localization of semantic information, and to highlight differences between real and artificial data (Valeriani et al., 2023; Tulchinskii et al., 2024; Cheng et al., 2023). Another approach to study the internal representation is to use topological methods of TDA. Studies on Convolutional Neural Networks (CNN) used topological descriptors to explore the shape of activation functions (Rathore et al.) or their relations to performance (Naitzat et al., 2020). Magai & Ayzenberg (2022) introduced persistent homology dimension as an estimator of the intrinsic dimension of internal representations in CNNs, while Barannikov et al. (2022) proposed a measure of similarity based on topological descriptors to compare representations. Betti numbers have been observed to

remain stable across different datasets for the same architectures and to decrease as depth increases (Suresh et al., 2023).

**Zigzag Persistence.** Zigzag persistence was introduced in (Carlsson & de Silva, 2010; Carlsson et al., 2009; Tausz & Carlsson, 2011) as an extension of persistent homology to study the persistence of topological features across sequences of spaces. This approach is particularly useful when data undergo dynamic changes or transformations over time. Since its introduction, zigzag persistence has been applied in various fields, including Hopf bifurcations in dynamical systems (Tymochko et al., 2020), commuting patterns in Great Britain's transportation network Myers et al. (2023), coral reef ecosystems (McDonald et al., 2023), cell location time series (Yang et al., 2023; Zhang et al., 2023), and honeybee aggregations (Gharooni-Fard et al., 2024). It has also inspired methodological extensions such as multidimensional persistence (Kim & Mémoli, 2021) and the development of formigrams and crocker stacks (Xian et al., 2022).

**Layer Pruning by Similarity in Large Language Models.** Among existing methods to reduce the size of neural networks, layer pruning has gained particular relevance in the context of LLMs. The first applications to BERT models (Fan et al., 2020; Zhang & He, 2020; Fan et al., 2021; Jha et al., 2024) inspired a long series of experiments employing similar techniques (Sajjad et al., 2023; Siddiqui et al., 2024; He et al., 2024; Zhang et al., 2024a; Kim et al., 2024; Zhang et al., 2024b). Many of these efforts base their methodology on similarity measures of internal representations, which have conveniently been summarized in a recent review (Klabunde et al., 2023). In this work, we consider (Gromov et al., 2024), which uses angular similarity, and (Men et al., 2024), which uses Block-Influence similarity, as a reference point for comparison.

## 3 METHOD

In this section, we introduce the zigzag persistence framework, which we use to analyze the internal representations of LLMs pre-trained with an autoregressive loss. These models typically receive an input sequence of $n$ tokens (often representing a sentence) embedded in a $d$-dimensional space. The input is transformed across the network layers without altering the embedding dimension. Due to the autoregressive nature of these models, the representation of the last token in a sequence captures information about the entire sequence and is used to predict the next. As a result, we choose to focus on the last token representation of each sequence at each layer. Thus, our point cloud is represented by last tokens embeddings, i.e. vectors of the form $\{\mathbf{x}_i(\ell_j)\} \in \mathbb{R}^d$, for $i = 1, ..., N_{\text{sentences}}$ and $j = 1, ..., N_{\text{layers}}$. These last tokens are extracted from large datasets of text and serve as an observational probe of the manifold we would like to model.

### 3.1 TOPOLOGICAL DATA ANALYSIS AND PERSISTENT HOMOLOGY

Topological data analysis (Edelsbrunner et al., 2002; Zomorodian & Carlsson, 2004) provides a tool for geometrically characterizing highly complex datasets. Within this framework, persistent homology (Carlsson, 2009) is the key methodology to characterize a point cloud on multiple scales at once. Its goal is to identify the range of scales over which a particular class of topological features (connected components, loops, voids, higher dimensional "holes") remain relevant, or "persistent", as opposed to "topological noise", i.e. features disappearing roughly at the same scale they formed. The basic ingredients for this technique are i) a criterion to connect points, forming a *simplicial complex* and ii) a scale parameter $\nu$ (often a coarsening scale) such that given $\nu_1 \leq \nu_2$, then the two corresponding simplicial complexes are related by $K_{\nu_1} \subseteq K_{\nu_2}$. The ordered sequence of simplicial complexes for varying scale parameters is called *filtration*. An intuitive example is the Vietoris-Rips filtration, built from complexes parametrized by the radius of the ball drawn around each point of the dataset.

Filtrations can be generalized to a more flexible structure called a *zigzag filtration*. Unlike a standard filtration, a zigzag filtration allows the sequence of complexes to move both forward and backward, meaning that inclusions between complexes can reverse at certain steps. We take this approach in our study to track the evolution of the internal representations *across* layers, rather than at a fixed snapshot, as done in traditional persistent homology implementations. In this sense, our parameter is not a distance/coarsening scale, but a discrete *time* scale represented by the layer number. We track topological features as they are formed and destroyed along the layers of the model and we statisti-

cally characterize these changes to describe a complex series of transformations in high-dimensional space. Differently than standard persistent homology, short- and long-lived features represent how the model dynamically evolves. Short-lived features indicate a high rate of rearrangement of the points $x_i$ between adjacent layers, while long-lived features suggest a phase of retention of (relative) positions in the model. This is a crucial point in our analysis, as it provides a novel tool to geometrically interpret the model's internal representations. We now outline the main steps of the zigzag algorithm, leaving a rigorous mathematical formulation to Appendix A.

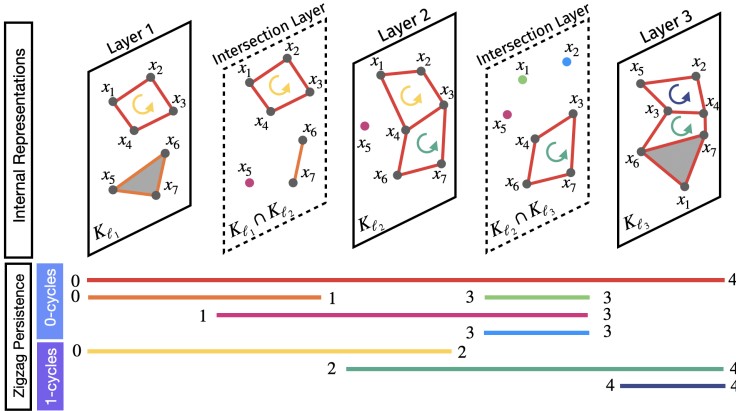

Figure 1: A schematic representation of the zigzag algorithm.

## 3.2 THE ZIGZAG ALGORITHM

We aim to study internal representations by tracking statistical changes in the formation of $p$-dimensional holes, or $p$-cycles, generated by connecting nearby data points within each layer $\ell_i$. As introduced above, the first ingredient for a TDA formulation is a criterion for connecting points of the dataset. In this regard, we construct a k-Nearest Neighbors graph $G_{\ell_i} = (V_{\ell_i}, E_{\ell_i})$ at every layer $\ell_i$, where the number $k_{\text{NN}}$ of neighbors is a fixed hyperparameter (see Le & Taylor (2024) for a previous use of a $k_{\text{NN}}$-based filtration). To exploit the knowledge that the manifold on which the data lie is typically much smaller than the high-dimensional ambient space, we extend the dimension of the graph by filling higher-dimensional simplices. More precisely, we *fill* a simplex when its boundary, composed of lower-dimensional simplices (such as vertices and edges), is complete. In particular, we consider a triangle as filled when it has three vertices with pairwise connections. Similarly, a tetrahedron is filled when four vertices are all interconnected by edges, totaling six edges. This concept extends to higher dimensions up to a specified maximum dimension $m$. Thus, in each layer, we construct the simplicial complex $K_{\ell_i}$ defined by:

$$K_{\ell_i} = \bigcup_{S \subseteq V_{\ell_i}} \left\{ S \mid \forall x_s, x_l \in S, \ (x_s, x_l) \in E_{\ell_i} \text{ and } |S| \leq m+1 \right\}. \tag{1}$$

To track changes in the network, we compute intersection layers by identifying simplices present simultaneously in both adjacent layers. This allows us to construct a sequence of inclusions between these complexes

$$K_{\ell_1} \swarrow \quad \nearrow K_{\ell_2} \swarrow \quad \nearrow K_{\ell_{L-1}} \swarrow \quad \nearrow K_{\ell_L} \tag{2}$$

$$K_{\ell_1} \cap K_{\ell_2} \qquad \dots \qquad K_{\ell_{L-1}} \cap K_{\ell_L}$$

where we define $L \equiv N_{\text{layers}}$ for conciseness. This sequence represents our zigzag filtration, denoted by $\Phi$. This filtration is the second ingredient needed to define persistent homology. We thus define a notion of *birth* and *death* of $p$-dimensional topological features, also denoted as $p$-cycles, with $p = 0, ..., m-1$, being $m$ the maximum dimension at which we expand the graph. Throughout this work, we choose $m = 4$, which implies that the $p$-cycles are well defined up to dimension $p = 3$. These cycles can be thought of as holes in their respective dimension. We can track the persistence

of these cycles as they appear in a given layer when a group of points exhibits a particular proximity and distribution in the complex and disappear at a subsequent layer when some points have moved apart, causing the cycle to vanish. We illustrate the idea in Figure 1. The output of the zigzag algorithm is then a multiset of birth-death pairs $[\text{birth}, \text{death}]$[1], known as the *persistence diagram*

$$\text{Pers}_{\text{p}}(\Phi) = \Big\{ [\text{birth}, \text{death}] \mid \text{birth}, \text{death} \in \{0, \dots, 2N_{\text{layers}} - 1\} \Big\}. \tag{3}$$

We thus work with a zigzag filtration naturally indexed by $\{0, 1, 2, \dots, 2N_{\text{layers}} - 1\}$. Specifically, as shown in the Figure 1, even numbers starting from $0$ are assigned to $p$-cycles that emerge and disappear within the model layers. In contrast, odd numbers are designated for features at the intersection layers. It is important to note that $p$-cycles are defined as equivalence classes, meaning that a cycle need not maintain the same form at the level of simplices throughout its lifetime. The orange $0$-cycle in the figure exemplifies this: in Layer 1, the cycle corresponds to a filled triangle, $\{x_5, x_6, x_7\}$, but in the intersection layer, it is reduced to the edge $\{x_6, x_7\}$. In Layer 2 this edge merges with another $0$-cycle (depicted in red), marking the death of the orange cycle. A mathematical explanation of this is provided in Appendix A. This feature ensures robustness of our construction to small changes in the $k_{\text{NN}}$ graph. The corresponding algorithm that generates $\text{Pers}_{\text{p}}(\Phi)$ is schematically described below.

---

**Algorithm 1** Zigzag algorithm

---

**Require:** $model, dataset, k_{\text{NN}}, m$
  $reps \leftarrow \text{extractRepresentations}(model, dataset)$
  $K \leftarrow []$
  **for** $i \leftarrow 1$ to $model.\text{getNumLayers}()$ **do**
    $graph \leftarrow \text{kNearestNeighborsGraph}(reps[i], k_{\text{NN}})$
    $K.\text{append}(\text{graphExpansion}(graph, m)$
  **end for**
  $K_{\text{int}} \leftarrow \text{computeIntersectionLayers}(K)$
  $f, times \leftarrow \text{computeFiltrationTimes}(K, K_{\text{int}})$
  $\Phi \leftarrow \text{FastZigZag}(f, times)$

---

It exploits two existing public codes that were developed for zigzag computations: DIONYSUS2 (Morozov) and FASTZIGZAG (Dey & Hou, 2022). DIONYSUS2 is a C++ library for computing persistent homology, with a specific library for zigzag persistence. In our case, it has the role of extracting the filtration $f$ and computing the $times$ array, i.e. the list of layer indices to be associated with the birth and death of features. FASTZIGZAG allows to calculate efficiently [2] the persistence diagram $\text{Pers}_{\text{p}}(\Phi)$ by converting the input zigzag filtration to a non-zigzag filtration of an equivalent complex with the same length, and it then converts the obtained persistence intervals back to zigzag.

### 3.3 Effective Persistence Image

The pairs generated within $\text{Pers}_{\text{p}}(\Phi)$ are best understood by visualizing them through a *persistence image*, a well-known descriptor within the TDA tools. The persistence image in our case results in a grid of size $(2N_{\text{layers}} - 1) \times (2N_{\text{layers}} - 1)$, for each homology dimension $p$. Each pixel in the grid is associated with an integer value corresponding to the number of cycles appearing with that birth-death pair. Defined this way, the persistence image does not discriminate between the model and intersection layers. Their behavior is generally fairly different, and have an alternating structure between model and intersection layers. Hence, persistence images are not *smooth* as a function of layers. To achieve a smoother representation, we introduce *effective persistence images*, obtained by excluding the intersection layers from the construction. This is achieved by defining a map, similar to the approach in (Kim & Mémoli, 2017), that translates the collection of intervals from the zigzag

---

[1]The repetition of a pair $[\text{birth}, \text{death}]$ indicates that multiple cycles in dimension $p$ have been created and destroyed in correspondence of the same layers.

[2]The algorithm performs well even for the relatively large datasets we employ for this analysis: with 10K points embedded in a space with dimension $d = 4096$, a number of neighbors for the $k_{\text{NN}}$ graph of $k_{\text{NN}} = 10$, and a maximum homology dimension of $m = 10$ on an AMD EPYC 7H12 it takes approximately 2 hours.

persistence diagram of the filtration in equation 2 into intervals, where the birth and death occur only across model layers. Formally, for $b, d > 0$, we obtain:

$$\widehat{PI}_p(b/2, d/2) = PI_p(b, d) + PI_p(b-1, d) + PI_p(b, d-1) + PI_p(b-1, d-1), \quad (4)$$

where $\widehat{PI}_p$ is the effective persistence image for the $p$-cycles and $b, d$ are model layers indexed by even numbers.[3] The collection of $\widehat{PI}_p$s taken over all $p$ contains all the information output from our zigzag algorithm, and give a useful overview of the model as a whole. On the other hand, they are not easily tractable in a statistical sense and hard to interpret. Indeed, one focus of this work is to look at the fine-grained topological structure of representation space, tracking the persistence of cycles across layers. For this purpose, we develop a suited summary of the effective persistence image in the next section.

### 3.4 PERSISTENCE SIMILARITY

Given two layers $\ell_1, \ell_2$, we define the *persistence similarity* [4] as the fraction of $p$-cycles in $\ell_1$ that exist in $\ell_2$ as well, and have existed throughout the layers in between. Mathematically it can be expressed as

$$\mathcal{S}_p(\ell_1, \ell_2) = \frac{\sum_{\ell_1 \leq M_1, \ell_2 > M_2} \widehat{PI}_p(\ell_1, \ell_2)}{\beta_p(\ell_1)} \quad (5)$$

$$M_1 = \min(\ell_1, \ell_2); \quad M_2 = \max(\ell_1, \ell_2)$$

where $\beta_p(\ell)$ is the Betti number, i.e. the number of alive $p$-cycles at layer $\ell$. [5] Given a $p$-cycle that is alive at a given layer $\ell$, we can thus define the average probability of finding it alive at any other layer as

$$\bar{\mathcal{S}}_p(\ell) = \frac{1}{N_{\text{layers}}} \sum_{\ell_i = 1}^{N_{\text{layers}}} \mathcal{S}_p(\ell, \ell_i), \quad (6)$$

which indicates the degree of "mobility" of the system at a given layer, i.e. overall retention of cycles in each model layer. Thus, a low value of $\bar{\mathcal{S}}_p$ represents a phase during which internal representations are undergoing major topological changes, causing points of the dataset to change relative positions abruptly. For high values, the inverse is true, i.e. the relations among points are relatively stationary. It is worth noting that traditional measures of similarity between layers typically depend solely on their current state, namely the activation matrices on the set of data. In contrast, our method considers the trajectory from $\ell_1$ to $\ell_2$, implying that persistence similarity does not just depend on the initial and final states but also on the path between them.

## 4 EXPERIMENTS

### 4.1 MODELS, DATASETS AND BENCHMARKS

We work with 4 models: Llama2 (Touvron et al., 2023), Llama3 (AI@Meta, 2024), Mistral (Jiang et al., 2023) and Pythia 6.9b (Biderman et al., 2023). These models are open-source decoder-only transformers, and they achieve high performance in the benchmarks we consider in this work. Llama2-7B, Llama3-8B, Mistral 7B, and Pythia 6.9b have 32 hidden layers, Llama2-13B has 40 hidden layers, and both Llama2-70b and Llama3-70b have 80 hidden layers.

For our purposes, the input dataset from which we take internal representations must provide a fair test of how the model processes and understands language. An extensive and accessible corpus is the Pile dataset (Gao et al., 2020), which combines 22 datasets over a wide range of topics and

---

[3]Note that this operation does not modify the information about the model layers contained in the original $\text{Pers}_p(\Phi)$, as it redefines consistently all the births and deaths.

[4]We note that the terminology "persistence similarity" has been used in previous literature in a different context and application Xia (2018). We thank a reviewer for providing us with this reference.

[5]Note that equation 5 is well-defined only when $\beta_p(\ell) > 0$. If there are no $p$-cycles at either $\ell_1$ or $\ell_2$, $\mathcal{S}_p(\ell_1, \ell_2)$ should be 0 by definition. We omitted this limit case from equation 5 for readability.

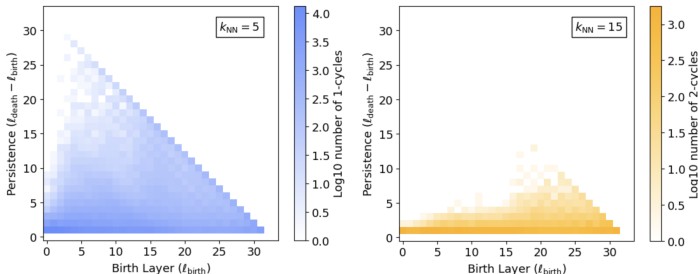

Figure 2: Effective persistence image for the Llama 3 8B model using the SST dataset. We show 1-cycles (Left Panel) and 2-cycles (Right panel). The corresponding $k_{\mathrm{NN}}$ graph is constructed with $k_{\mathrm{NN}} = 5$ and $k_{\mathrm{NN}} = 15$, respectively. The density plot shows the amount of cycles (colorbar) for a given birth-persistence pair (x- and y-axis), where values refer to the model layer.

structures. For computational reasons, we take the Pile-10k subset, accessible on HugginFace.[6] For completeness, we also consider the Standford Sentiment Treebank (SST) dataset (Socher et al., 2013). From these datasets, each prompt is processed so that the last token is extracted at each normalization layer and the final normalization applied to the output layer.

We use 3 benchmarks for layer pruning performance evaluation: MMLU (Hendrycks et al., 2021), HellaSwag (Zellers et al., 2019), and Winogrande (Sakaguchi et al., 2019), which have been widely used for similar purposes in previous analyses. The benchmarks are evaluated for the models with the use of the library lm-eval-harness by (Gao et al., 2024) with a 5-shot setup.

## 4.2 Zigzag persistence applied to LLM models

**Effective Persistence Image.** We generate an effective persistence image for each model using the two datasets, each homology dimension up to $p = 3$, and for a range of values of $k_{\mathrm{NN}} \in [1, 15]$. We show an example of this effective persistent image in Figure 2 for the Llama 3 8B model for the SST dataset for 1- and 2− cycles for $k_{\mathrm{NN}} = 5$ and $k_{\mathrm{NN}} = 15$, respectively.[7] The choice of the hyperparameter $k_{\mathrm{NN}}$ is done so as to maximize the total number of cycles. The x-axis represents the layer at which a $p$-cycle is born, and the y-axis represents persistence, i.e. death layer - birth layer. The colorbar measures the amount of $p$-cycles on a given grid point. As expected, a large number of cycles are very short lived, i.e. the grid points at persistence equal unity. On the other hand, we observe that persistence is typically higher for $p$-cycles born after the first half of the model's depth, a feature that is visually evident on the right panel of Figure 2, representing 2-cycles, but observed across all models and dimensions, especially for 1-cycles. A fraction of these cycles have maximal persistence, i.e. they survive until the last layer.

In computing $\widehat{PI}_p$s across models, dimensions and $k_{\mathrm{NN}}$ values, we observe that 0- and 3-cycles are relatively low in number, while 1- and 2-cycles are higher, reaching tens of thousands of cycles per layer. This behaviour might be expected for a $k_{\mathrm{NN}}$-graph based costruction, since connections are dense even for low values of $k_{\mathrm{NN}}$, especially if points are concentrated in low dimensional regions of the representation space. We examine this behavior in detail to make sure that our construction is stable for different choices on the $k_{\mathrm{NN}}$ graph, see Appendix B for details.

The $\widehat{PI}_p$s from Figure 2 are suggestive of important features in the topology of internal representations, which we look at in more detail using persistence similarity. Given the prevalence of 1-cycles across various models, layers, and choices of $k_{\mathrm{NN}}$, we concentrate on these features in the following discussion.

**Persistence similarity.** We can visualize persistence similarity, as defined in equation 5, $\mathcal{S}_1(\ell_1, \ell_2)$ as a density plot, shown in Figure 3 for Llama 3 of two different sizes (8B and 70B) and the SST

---

[6]https://huggingface.co/datasets/NeelNanda/pile-10k

[7]The reason for using the SST dataset, instead of Pile, is that the 70B models are computationally expensive for the latter. We show that results are in agreement between the two models in Appendix C.

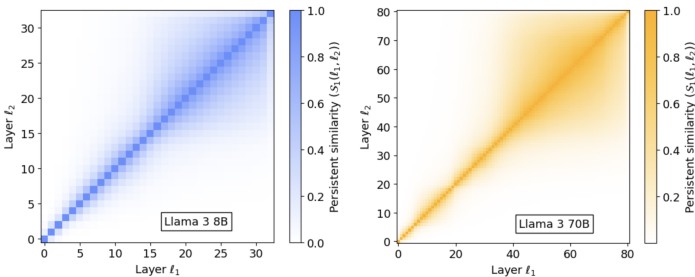

Figure 3: Persistence similarity of 1-cycles as defined in equation 5 for Llama 3 of two different sizes (8B and 70B) and the SST dataset. For both models we fix $k_{NN} = 5$. A given pixel of the grid represents similarity computed between two layers. Darker regions indicate higher similarity.

dataset, with corresponding plots for Pile for the 8B shown in Appendix C. Again, we choose $k_{NN} = 5$ although results are similar within the $k_{NN} \in [1, 15]$ range. In these plots, darker regions indicate a higher fraction of $p$-cycles alive between two given layers. Note that the plot is not symmetric by definition (cfr. equation 5) meaning that at a given layer, the fraction of cycles alive at an earlier layer might be different than the ones alive at a later layer. Nevertheless, $\mathcal{S}_p$ is approximately symmetric. Both models clearly show a high degree of similarity roughly midway through the depth, until before the last few layers. This is in agreement with what observed in the $\widehat{PI}_p$, which suggested that a $p$-cycle born after the first half of the model is likely to survive until the last layer. We now compute

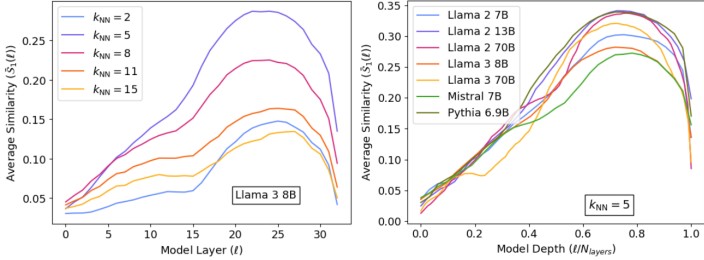

Figure 4: Average Similarity as a function of model depth, computed for Llama 3 8B and varying $k_{NN}$ parameter (Left Panel) and at fixed $k_{NN} = 5$ parameter and varying models (Right Panel).

the average similarity, i.e. the average over the column of persistence similarity (cfr. equation 6), $\bar{\mathcal{S}}_p$ both at fixed model and varying $k_{NN}$ parameter, and at fixed $k_{NN}$ parameter and varying model, as a function of the model's depth. Results are shown in the Left and Right Panels of Figure 4, respectively. Based on the Left Panel, we choose $k_{NN} = 5$ as representative value for the Right Panel, given that it gives the highest values of $\bar{\mathcal{S}}_1$. Remarkably, $\bar{\mathcal{S}}_1$ peaks at the same relative depth for a wide variety of models, while the parameter $k_{NN}$ only changes the normalization of the curve.

Overall, we can identify three distinct phases:

- **An increasing phase**, lasting from early to middle layers. During this phase, the rate of increase of similarity is constant, and seemingly universal, i.e. it does not depend on the nature of the model, its size and very weakly on the dataset used (cfr. Figure 7 (right panel) in the Appendix C). It does depend on the underlying filtration (cfr. Figure 4 (left)). [8] The positive rate of increase suggests that the average persistence of cycles is growing, indicating that transformer architectures are gradually retaining more and more features from the dataset;

- **A plateau phase**, during which average similarity saturates to a global maximum;

- **A decreasing phase**, in the last few layers of the model. During this phase, features are progressively destroyed and are increasingly unlikely to persist long.

---

[8] We have verified that for different values of $k_{NN}$, the universality of the increase across models is conserved.

We deserve a more detailed analysis of these phases and their implications for model behavior for Appendix D.

### 4.3 LAYER PRUNING BY PERSISTENCE SIMILARITY

Recently, measures of layer similarity have been used to identify layers that contribute minimally to the performance of LLMs. These layers can be pruned, and the performance re-evaluated to validate this assumption. Since persistence similarity tracks changes across layers, it can be leveraged for layer pruning by selecting layers that retain the most cycles. Consequently, we establish a pruning criterion based on average persistence similarity $\bar{\mathcal{S}}_1$ computed now on the Pile dataset. Specifically, we prune layers that lie within $10\%$ and $20\%$ of the maximum $\bar{\mathcal{S}}_1$, corresponding to conservative and aggressive pruning, respectively. Here is a schematic summary of the algorithm.

---

**Algorithm 2** Pruning algorithm

---

**Require:** $\bar{\mathcal{S}}_1, model, threshold,$
    $max \leftarrow \max(\bar{\mathcal{S}}_1)$
    $layersToRemove \leftarrow []$
    **for** $l \leftarrow 1$ to $model.getNumLayers()$ **do**
        **if** $\bar{\mathcal{S}}_1[l] > max * threshold$ **then**
            $layersToRemove.append(l)$
        **end if**
    **end for**
    $model.removeLayers(layersToRemove)$

---

The algorithm outputs how many and which layers have high degree of persistence similarity. We now cut those layers and measure performance using the benchmarks introduced in Section 4 and across models considered in this work.

We compare to layer pruning methods based on state of the art measures of similarity, namely (Gromov et al., 2024) and (Men et al., 2024). Both approaches are designed to take as input the desired number of layers to prune $N_{\text{prune}}$ and measure performance as $N_{\text{prune}}$ grows. For a fair comparison, we feed the number of layers cut by our method as an input to the other two methods, and verify which layers they select to cut given this input, and the corresponding performance. We show a schematic diagram of the layers cut with our method (Bottom Row) and the other two methods (Upper Row) in Figure 5. Interestingly, both considered methods from (Gromov et al., 2024) and (Men et al., 2024) give the same result at fixed $N_{\text{prune}}$, thus we refer to them simply as "other works".

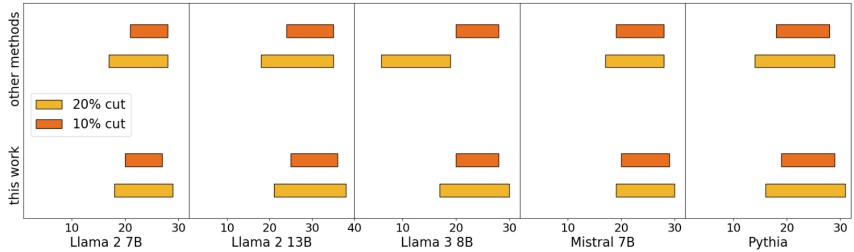

Figure 5: Pruned layers across models based on persistence similarity (Bottom Row) and other methods from (Gromov et al., 2024; Men et al., 2024). Since both these two methods give the same results, we generically call them "other works". The number of layers pruned for all methods is defined by cutting layers that are within $10\%$ (orange) and $20\%$ (yellow) of the maximum average similarity, $\bar{\mathcal{S}}_1$.

The $10\%$ pruning is rather stable across methods, with small variations. The more aggressive cut of $20\%$ generates more discrepancies, especially for Llama 3 8B, where both methods from (Gromov et al., 2024) and (Men et al., 2024) prefer to cut earlier layers.

| Models | MMLU | | | HellaSwag | | | WinoGrande | | |
|---|---|---|---|---|---|---|---|---|---|
| | Full | This work | Other works | Full | This work | Other works | Full | This work | Other works |
| Llama 2 7B | 45.74 | 37.38 (39.32) | **43.95** (34.35) | 58.54 | **44.71** (32.10) | 42.78 (35.10) | 74.43 | **68.67** (59.67) | 67.72 (62.67) |
| Llama 2 13B | 54.60 | 50.16 (36.45) | **50.71** (37.91) | 61.43 | **48.60** (34.35) | 47.84 (34.52) | 76.72 | 71.67 (63.21) | **73.15** (61.47) |
| Llama 3 8B | 65.07 | **53.44** (23.16) | **53.44** (24.33) | 61.37 | **41.60** (29.69) | **41.60** (27.10) | 77.10 | **70.00** (59.75) | **70.00** (50.58) |
| Mistral 7B | 62.40 | **53.17** (24.26) | 38.20 (37.86) | 62.83 | **36.67** (26.26) | 34.45 (28.10) | 77.35 | **66.50** (57.76) | 63.76 (55.96) |
| Pythia | - | - | - | 49.70 | 31.43 (31.23) | **34.96** (26.84) | 63.30 | 55.71 (54.84) | **58.09** (51.07) |

Table 1: **Benchmark Table.** For each benchmark we show three columns: (i) *Full*, represents the accuracy of the model without any layer pruned. (ii) *This work*, accuracy of the model with two different cuts, at 10% and 20%, where layers are pruned following the algorithm 2). The results are in the form 10% cut (20% cut) (iii) *Other works*, accuracy obtained by considering the same amount of layer pruned estimated with our method and then computing the layer to be pruned with two different similarity measures: angular distance from (Gromov et al., 2024) and Bi-score from (Men et al., 2024). The chosen layers turn out to be the same for the two methods, so the results are condensed in one column, and they are then represented in the format *first-block-cut*(*second-block-cut*).

We now show performance results in Table 1, [9] where in bold we indicate the layer pruning method that has better or equal performance with respect to the other method. Despite often selecting different layers, our topology-based pruning strategy achieves comparable results to methods from (Gromov et al., 2024) and (Men et al., 2024). We further test how much performance changes with pruning layers at different model's depths in Appendix D.

## 5 CONCLUSIONS

In this study, we present an innovative framework that utilizes zigzag persistence, a tool from Topological Data Analysis (TDA), to examine the internal representations of Large Language Models (LLMs). By employing various datasets as observational probes of the manifold on which the model functions, we aim to offer an interpretable depiction of changes in position and relationships across layers. A distinguishing feature of our framework is its ability to trace the emergence and disappearance of topological features as they evolve across layers. This approach effectively models the transformer architecture as an evolving dynamic system, setting it apart from previous research. With this algorithm, we introduce a new topological descriptor, persistence similarity, which statistically models rearrangements of points in representation space, and the rate of these changes, across layers. As a showcase experiment, we prune layers by identifying the ones with highest similarity and verify that this operation does not significantly compromise performance, yielding results comparable to state-of-the-art methods. Persistence similarity shows stability under models, datasets, and hyperparameters changes suggesting a universal topological structure in LLM representations.

There are several limitations in our study that future research could address. First, while our method shows robustness across hyperparameters within the framework, these choices need not be optimal. Defining an appropriate criterion for connecting points in the representation space, and consequently, a filtration, is a delicate task in TDA that could require further investigations to detail the impact of the various choices on the construction of the filtration. The information content of persistence similarity on internal representations and model behavior has not been investigated in detail (but see a few experiments in Appendix D) and it certainly deserves further investigation. Lastly, our study primarily focuses on static, pre-trained models. Extending this framework to track the evolution of internal representations during training would require computational optimization of the algorithm but could provide useful insights on model efficiency and behavior.

---

[9] Results for Pythia on MMLU tasks are not shown because the model is not designed for following the format of the tasks, as shown in (Biderman et al., 2023).

## 6 REPRODUCIBILITY

All the results contained in this work are reproducible by means of an anonymised repository that can be found at this link: https://anonymous.4open.science/r/conferenceProject-019A/.

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

## A   MATHEMATICAL FORMULATION OF ZIG ZAG PERSISTENCE

Zigzag persistence is a computational topology method that extends classical persistent homology to handle more complex data structures and filtration processes. Unlike standard persistence, which analyzes a single sequence of spaces filtered by inclusion, zigzag persistence allows for the exploration of data where sequences of spaces and maps can move both forward and backward.

A *zigzag filtration* of topological spaces is a sequence:

$$\chi\colon \mathbb{X}_1 \longleftrightarrow \mathbb{X}_2 \longleftrightarrow \cdots \longleftrightarrow \mathbb{X}_{n-1} \longleftrightarrow \mathbb{X}_n, \tag{7}$$

where each $\mathbb{X}_i$ is a topological space and each arrow $\longleftrightarrow$ represents a continuous function pointing forwards $\mathbb{X}_i \longrightarrow \mathbb{X}_{i+1}$ or backwards $\mathbb{X}_i \longleftarrow \mathbb{X}_{i+1}$.

If we apply a homology functor $H_p$ with coefficients in a field $\mathbf{k}$ to such a filtration, we get a zigzag filtration of $\mathbf{k}$-vector spaces, called *zigzag module*:

$$H_p(\chi)\colon H_p(\mathbb{X}_1) \longleftrightarrow H_p(\mathbb{X}_2) \longleftrightarrow \cdots \longleftrightarrow H_p(\mathbb{X}_{n-1}) \longleftrightarrow H_p(\mathbb{X}_n). \tag{8}$$

It is proven in (Carlsson & de Silva, 2010) that the algebraic classification of zigzag modules resembles Gabriel's classification of the persistence module described in (Gabriel, 1972). In particular, every finite-dimensional zigzag module, i.e. for which all the $\mathbf{k}$-vector spaces in the sequence that are finite-dimensional, can be decomposed as a direct sum of interval modules, where a (finitely indexed) *interval module* is a module of the form:

$$\mathcal{I}_{[b,d]}\colon I_1 \longleftrightarrow I_2 \longleftrightarrow \cdots \longleftrightarrow I_n, \tag{9}$$

where $I_i = \mathbf{k}$ for $b \leq i \leq d$, and $I_i = 0$ otherwise, and every arrow of the form $\mathbf{k} \longleftarrow \mathbf{k}$ or $\mathbf{k} \longrightarrow \mathbf{k}$ is the identity map. Moreover, the list of summands is unique up to reordering.

The *zigzag persistence diagram* of a filtration $\chi$ in dimension $p$ is the multiset of intervals $[b, d]$ corresponding to the list of interval summands $\mathcal{I}_{[b,d]}$ of $H_p(\chi)$. In other words,

$$\mathrm{Pers}_p(\chi) = \{[b_j, d_j]\colon j \in J\} \Longleftrightarrow H_p(\chi) \cong \bigoplus_{j \in J} \mathcal{I}_{[b_j, d_j]} \tag{10}$$

Each interval $[b, d]$ is called *persistence interval* and is thought of as a persistent homological feature of $\chi$ that appears at time $b$ (referred to as the "birth") and disappears at time $d$ (referred to as the "death[10]").

---

[10]In our setting we say a $p$-cycle "dies", we mean that the corresponding homology class no longer persists in subsequent layers. In the zigzag filtration, this happens when the cycle is no longer represented by an independent equivalence class in the homology group.

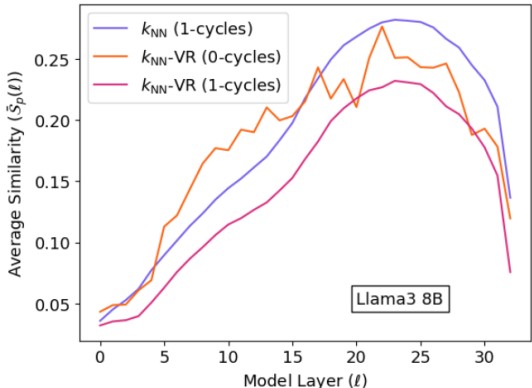

Figure 6: Plot of the Average Similarity as a function of model layers computed for Llama3 8B for both $k_{\text{NN}}$ and $k_{\text{NN}}$-VR complexes. We impose the number of 0-cycles, $\beta_0 = 500 \pm 100$ to build the $k_{\text{NN}}$-VR complexes.

In our approach, the use of intersection layers is essential for computing zigzag persistence, as it allows the construction of injective maps between the $k_{\text{NN}}$ complexes of model layers (see equation 2)[11]. Since our primary goal is to analyze the topological changes between model layers, we eliminate the construction of intersection layers while preserving the topological features by shifting each persistence interval such that the birth and death times occur strictly within the layers.

For an interval $[b, d]$ in the zigzag persistence diagram of dimension $p$ of filtration 2, the mapping that enables a bijective transformation to a new interval $[\hat{b}, \hat{d}]$[12] only across model layers is defined as follows:

$$\hat{b} = \begin{cases} b+1 & \text{if } b \text{ is an intersection layer} \\ b & \text{otherwise} \end{cases} , \quad \hat{d} = \begin{cases} d+1 & \text{if } d \text{ is an intersection layer} \\ d & \text{otherwise} \end{cases} \tag{11}$$

The relationship between the persistence image and the effective persistence image for $p$-cycles, denoted respectively by $PI_p$ and $\widehat{PI}_p$, where $b, d$ are the model layers indexed by even numbers, is described by the following system of equations:

$$\begin{cases} \widehat{PI}_p(0,0) = PI_p(0,0) \\ \widehat{PI}_p(b/2, d/2) = PI_p(b,d) + PI_p(b-1,d) + PI_p(b,d-1) + PI_p(b-1,d-1) \\ \widehat{PI}_p(b/2, \infty) = PI_p(b,\infty) + PI_p(b-1,\infty). \end{cases} \tag{12}$$

---

[11] An alternative method for constructing these maps and obtaining the zigzag persistence diagram is to use a filtration where, instead of intersections, the union of the complexes from two consecutive layers is considered. However, the Diamond Lemma, as discussed in (Carlsson et al., 2009), guarantees that both the intersection- and union-based filtrations encode the same homological information.

[12] By construction, all resulting intervals contain even numbers, as the model layers are indexed with these numbers.

## B  COMBINING THE $k_{\mathrm{NN}}$ GRAPH WITH THE VIETORIS-RIPS COMPLEX

The k-Nearest Neighbors ($k_{\mathrm{NN}}$) complex is built by expanding the corresponding $k_{\mathrm{NN}}$ graph to a fixed dimension $m$. A key limitation of the $k_{\mathrm{NN}}$ complex is that it ranks points by proximity without considering their actual distances. As a result, once $k$ is fixed on each layer, each point is connected to its k-Nearest Neighbors, regardless of the absolute distances involved. In our setting, the number of 0-cycles (the Betti [13] number $\beta_0$) of the $k_{\mathrm{NN}}$ complexes as a function of the layers tends to be unity, i.e. the whole complex is connected, even for relatively small values of $k_{\mathrm{NN}} \gtrsim 6$. This implies that 0-cycles contain no useful topological information on the internal representations.

To address this issue, we follow the approach in (Naitzat et al., 2020), which combines the $k_{\mathrm{NN}}$ complex with the Vietoris-Rips complex. Starting from the $k_{\mathrm{NN}}$ graph, the idea is to introduce a threshold radius $R$ on each layer and use it to filter out edges of the graph whose lengths are less than or equal to $R$, and then expand, denoting this new complex $k_{\mathrm{NN}}$-VR. This filtering step allows us to focus on longer-range connections, uncovering significant topological features that may be hidden by shorter, more local connections.

To ensure consistency across layers, we select the radius $R$ in each layer such that the number of 0-cycles, $\beta_0$, of the $k_{\mathrm{NN}}$ complex falls in a pre-determined range. We then compute the observables presented in this work and verify the results. For clarity, we refer to $k_{\mathrm{NN}}$ complex the construction used in the main body, and $k_{\mathrm{NN}}$-VR complexes the one presented in this section. For the sake of conciseness, we present only results for the average similarity $\bar{\mathcal{S}}$. In Figure 6 we show the average similarity of 1-cycles of the $k_{\mathrm{NN}}$ and the $k_{\mathrm{NN}}$-VR complexes and the 0-cycles of the $k_{\mathrm{NN}}$-VR complexes computed by imposing $\beta_0 = 500 \pm 100$. [14] We observe all three curves are qualitatively and quantitatively similar. This indicates that information about the similarity of 1-cycles remains unchanged, even when removing a considerable amount of short edges. Moreover, we observe the same information also on 0-cycles, now that we modified the complex such that their statistics are large enough to reliably compute similarity. We argue this indicates a universal (in homology) tendency to retain relational connections among particles in the middle-late layers of the model.

## C  CONSISTENCY OF RESULTS

To show the consistency of our method, we computed our observables on both representations from the Pile-10K dataset and SST dataset. For Pile-10K, we did not compute them on the largest models of 70B parameters to reduce computational usage. Nevertheless, we show here the effective persistence image, persistence similarity and average similarity in Figure 7.

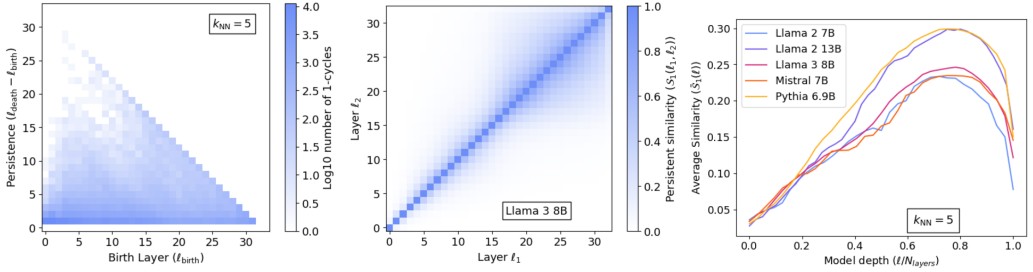

Figure 7: Effective persistence image (left), persistence similarity (middle) and average similarity (right) for the Pile-10K dataset.

---

[13] Betti numbers have been used in previous works (Naitzat et al., 2020; Suresh et al., 2023) for interpreting internal representations of neural networks. However, they describe each layer independently from the others, which is not the purpose of this work.

[14] We checked that results are stable as long as $\beta_0$ is much lower than the total number of points.

# D    IN-DEPTH ANALYSIS OF AVERAGE SIMILARITY

In this section, we perform experiments supporting a deeper understanding of the average similarity topological descriptor, and its implication for the model's behavior.

## D.1    AVERAGE SIMILARITY ON MATH AND CODE DATASETS

To assess if our method is sensitive to specialized datasets, we compute average similarity on three different datasets: SST, Math-12K[15] and Code-10K [16] from HuggingFace. The Math-12k dataset contains around 12K mathematical problems from different subfields of mathematics, while the Code dataset contains 115M files of code from Github from which we selected the first 10K. We

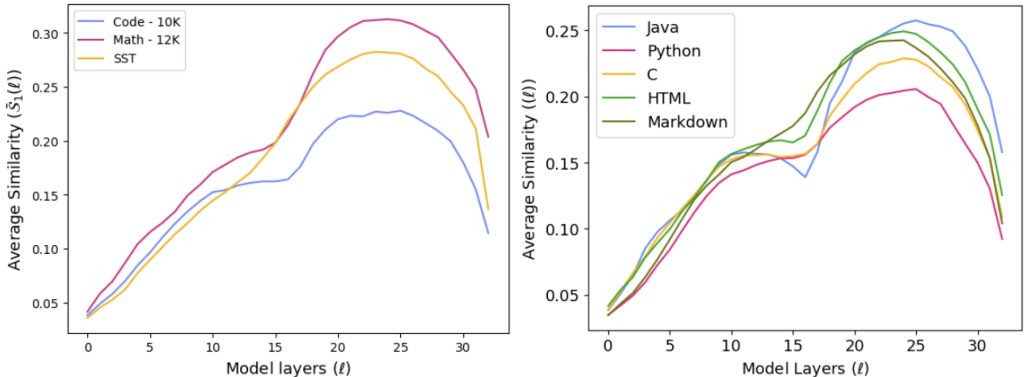

Figure 8: Plots of Average similarity for Llama 3 8B on SST, Code-10k, Math-10k (Left Panel), and on 5 different programming languages (Right panel). Each programming language is a dataset composed of 10K prompts. The ZigZag for both plot is run with $k_{\mathrm{NN}} = 5$.

present our results in Figure 8. In the left panel, we show that the increasing phase for both code and math datasets is split into two, with a previously unseen plateau in the middle. We argue that this behavior is triggered by special characters generally not used in conventional human language. We confirm this expectation by comparing different levels of verbosity in programming languages in the right panel of Figure 8: we see that the splitting of the phase is correlated with verbosity of the language (e.g markdown shows no split, C shows two distinct phases).

## D.2    SHUFFLING TEST

To test the plateau phase seen in average similarity across models, we perform a shuffling of tokens within the prompts of the SST and math dataset, as a way of destroying the structure and semantic coherence of the prompts, without modifying their unigram frequency distribution (see e.g. Cheng et al. (2024) for an application of shuffling to internal representations of transformers).

In Figure 9, we show how the plateau is modified by this change across two different datasets: the increase phase is shorter and the plateau is much lower in similarity.

## D.3    PERFORMANCE AND SIMILARITY

We can test the three phases also by pruning blocks of adjacent layers with a sliding window and testing the model on a benchmark. The scope of this experiment is to show how the phases seen in average similarity are linked to model performance. In Figure 10, we show performance of the MMLU benchmark against blocksizes of 5, 3 and 2 adjacent layers with sliding windows of 2, 1 and 1 for the left, middle and right panels, respectively. We see that performance is at the level of random choice during the increasing phase and it maximizes close to the maximum average similarity during the plateau phase. As an interesting finding, we see a drop in performance right in correspondence

---

[15]https://huggingface.co/datasets/lighteval/MATH
[16]https://huggingface.co/datasets/codeparrot/github-code

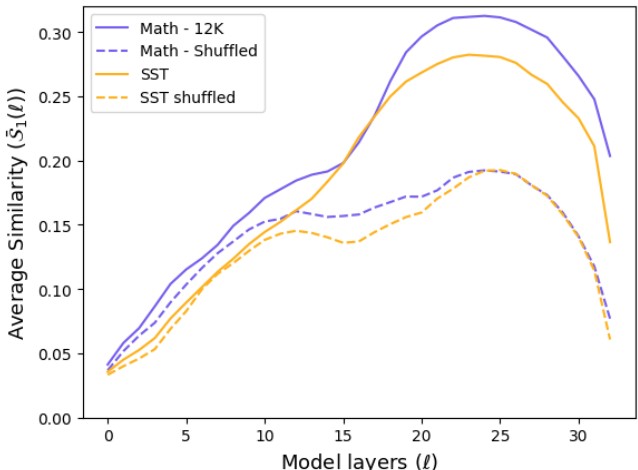

Figure 9: Plots of Average similarity for Llama 3 8B on SST, Code-10k, Math-10k (Left Panel), and on 5 different programming languages (Right panel). Each programming language is a dataset composed of 10K prompts. The ZigZag for both plot is run with $k_{NN} = 5$.

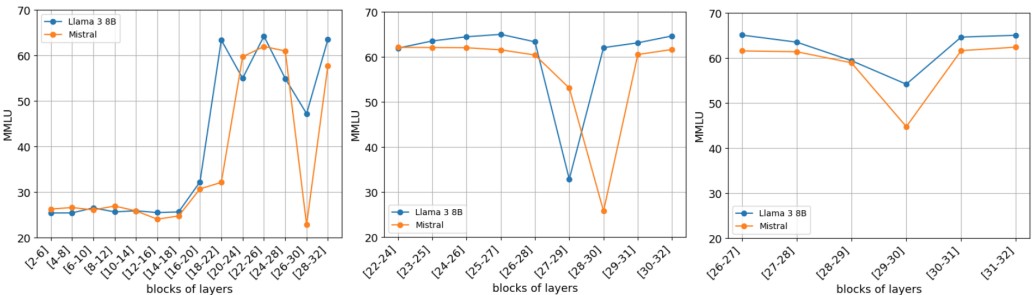

Figure 10: MMLU 5-shot benchmark run on Llama3 8B and Mistral. The different benchmarks showed are done by cutting blocks of layers with a fixed size and by changing the starting point with a sliding window. Left plot is made with a block size of 5 and sliding windows of 2, Center plot with a block size of 3 and sliding windows of 1, right plot with a block size of 2 and sliding window of 1.

of the decreasing phase. For both Llama and Mistral, the relevant layers are a few layers before the last. This finding deserves a closer investigation, which we leave for future work.

As a summary of these findings, in Figure 11 we plot average similarity for Llama (left) and Mistral (right) for three datasets (Math, Code and SST), where we highlight the end of the increasing phase, corresponding to an increase of performance when layer pruning and the beginning of the decreasing phase, corresponding to a sudden decrease of performance.

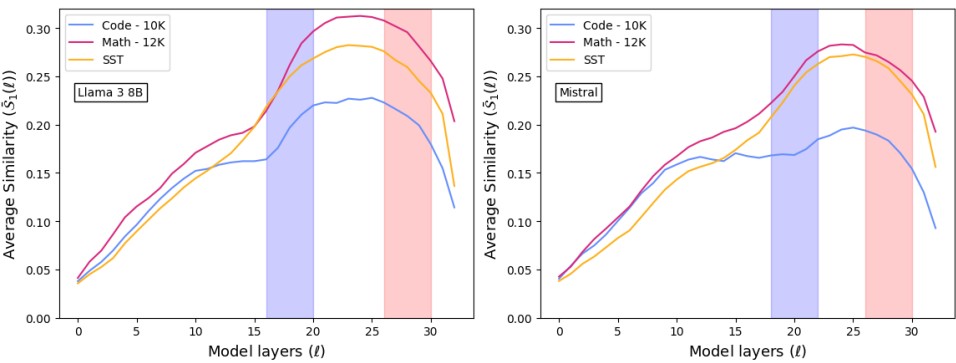

Figure 11: Average similarity for Llama3 8B (Left plot) and Mistral (Right plot) on three different datasets (Math-12K, Code-10K and SST), in blue are highlighted the last block layers with low performance of Figure10, while in red are highlighted the layers towards the end of the model where there is a local minima.

