# OpenReview forum: "Persistent Similarity in Internal Representations of Large Language Models"
_ICLR.cc/2025/Conference — Submitted to ICLR 2025_

### Official Review · Reviewer_TVBg · 2024-10-31

**Soundness:** 3
**Presentation:** 4
**Contribution:** 3
**Rating:** 5
**Confidence:** 3

**Summary:**

The paper proposes a new framework based on zigzag persistence to analyze internal
representations in LLMs by characterizing the birth and death of topological features within the
model’s layers.   it provides a fine-grained geometric analysis of the internal representations through TDA.  Distinct from traditional methods that solely compare representations at individual layers, the proposed method  captures their entire evolutionary path, providing a richer understanding of how these features evolve and contribute to the model’s decision-making processes

**Strengths:**

Originality:
The idea of applying topological data analysis to LLM  is novel and interesting
Clarity:
The paper is well-written and easy to follow.
Significance:
The proposed Persistence Similarity and zigzag algorithm have rigorous mathematical formulation and are easy to compute. The experiment in Section 4.3 is promising. Author compares the performance of layer pruning Persistence Similarity and other common llm layer pruning methods and the proposed method performs well in most cases under 10% pruning rate.

**Weaknesses:**

Major concerns:
1. The discussion in Section 4 did not clearly explain the sensitivity of the proposed method when pruning LLM layers.  The result shown in Fig 5 indicates that, across different models, our metric consistently identifies the deep layers as “redundant”. Additionally, in the results from Table 1, the proposed metric does not show an advantage at a 20% sparsity level, which further heightens my concerns.

2. The authors mention, "Due to the autoregressive nature of these models, the representation of the last token in a sequence captures information about the entire sequence and is the only token used for predicting the next." I disagree with this assumption; predicting the next token is clearly based on all preceding tokens. Therefore, I would like to see an analysis of certain special tokens, such as the EOS and BOS tokens.

3.  Can Persistence Similarity and the zigzag algorithm help explain the varied responses of LLMs across different abilities or areas of knowledge? The pruning experiments presented in the paper are limited to base models. Could you provide an analysis of the instruction-following abilities of chat models or their capabilities in math and code?

**Questions:**

Overall, I believe this paper is likely to present a novel perspective on understanding LLMs.
However, the experimental section is overly limited. I look forward to discussing these questions further in the weaknesses section. Addressing them may help improve the overall score.

---

> ### Author Response · Authors · 2024-11-26
> **Reply to Reviewer TVBg**
>
> ## Reply to TVBg
> > The discussion in Section 4 did not clearly explain the sensitivity of the proposed method when pruning LLM layers. The result shown in Fig 5 indicates that, across different models, our metric consistently identifies the deep layers as “redundant”. Additionally, in the results from Table 1, the proposed metric does not show an advantage at a 20% sparsity level, which further heightens my concerns.
>
> We thank the referee for raising a fair point. We reiterate that the goal of our work is *not* to provide with a new, competitive, method for layer pruning. We recognise this point might not have been clear, and we tried to clarify this further in the text. Nevertheless, we have expanded the discussion over the pruning methodology by running an experiment showing that performance is correlated with different phases in our metric, see Figure 10 and 11.
>
> > The authors mention, "Due to the autoregressive nature of these models, the representation of the last token in a sequence captures information about the entire sequence and is the only token used for predicting the next." I disagree with this assumption; predicting the next token is clearly based on all preceding tokens. Therefore, I would like to see an analysis of certain special tokens, such as the EOS and BOS tokens.
>
> We believe that the reviewer might have misunderstood the sentence, so we tried to rephrase it. The approach of considering only the last token for geometrical investigations of the internal representations has already been used by other works.
>
> > Can Persistence Similarity and the zigzag algorithm help explain the varied responses of LLMs across different abilities or areas of knowledge? The pruning experiments presented in the paper are limited to base models. Could you provide an analysis of the instruction-following abilities of chat models or their capabilities in math and code?
>
> We thank the referee for the questions that inspired us into making further experiments, which we explain in the general response and in Appendix D. Besides the new experiments, it might be useful to notice that our approach is based on the study of the manifolds on which an ensemble of prompts lie in, so we do not think that our measure is suitable for analyzing instruction-following abilities of single prompts, at least not as it is defined in our work. A way to address this suggestion would be to apply our method to sequences of tokens within the same prompts. An experiment in this direction would take us too far from the purpose of the present work, thus we deserve it to future work.

---

### Official Review · Reviewer_99nQ · 2024-11-01

**Soundness:** 2
**Presentation:** 2
**Contribution:** 2
**Rating:** 5
**Confidence:** 3

**Summary:**

This paper proposes a novel TDA-based method for measuring LLM similarity.  Zigzag-persistence is introduced to take account of the properties of the LLM graph structure. As part of the evaluation, it has been applied to Pruning to check its effectiveness.

**Strengths:**

- This paper introduces a new idea to introduce Zigzag-persistentce by considering the layer structure.
- It has been applied to pruining to check its effectiveness.

**Weaknesses:**

- Methods that regard Neural Networks as graphs and apply TDA have been proposed in [1], for example. This is the usual persistent homology, but the use of zigzag persistence for graph structures has been verified in [2],[3] and elsewhere. The structure focusing on the LAYER structure has some aspects of novelty, but it seems to be a simple conventional combination idea and the degree of novelty is weak.

[1] T. Lacombe et al., Topological Uncertainty: Monitoring trained neural networks through persistence of activation graphs, IJCAI2021

[2] A. Myerset al., Temporal network analysis using zigzag persistence. EPJ Data Science, 12(1):6, 2023.

[3] Y. Chen et al., Time-Conditioned Dances with Simplicial Complexes: Zigzag Filtration Curve based Supra-Hodge Convolution Networks for Time-series Forecasting NeuRIPS2022

- It is difficult to see what the similarity of the models is intended to achieve. The proposed method is similarity of the graph structure and may be similar for different models. The paper evaluates the similarity in application to pruning to demonstrate its validity. However, it is difficult to accept from the experimental results alone that the proposed method is a superior similarity simply because the implications that the similarity indicates are not clear. As a method for measuring the degree of change, such as model modifications, such as pruning, it is intuitively effective. However, as no uses have been indicated, the current situation is that there is only evidence for it as a pruning method. If the claim is that it is a superior pruning method, it should be clearly stated as such. In addition, if the pruning method is slaughtered and claimed, it should be compared with a generally sufficient advanced puruning method, for example [4],[5].

[4] S. Ashkboos et al., SliceGPT: Compress Large Language Models by Deleting Rows and Columns, ICLR2024

[5] X. Ma et al.,LLM-Pruner: On the Structural Pruning of Large Language Models, NeuRIPS2023

**Questions:**

- Clarify the claim point(s) for which the novelty of the contribution in this paper is sufficient.
- Please clearly indicate in what sense the similarity of the proposal is valid, what use scenarios are covered (other than pruning) and whether it is sufficient evidence for this.


Why are the titles on the system different from the titles in the PDF?

---

> ### Author Response · Authors · 2024-11-26
> **Reply to Reviewer 99nQ**
>
> ## Reply to 99nQ
>
> > Methods that regard Neural Networks as graphs and apply TDA have been proposed in [1], for example. This is the usual persistent homology, but the use of zigzag persistence for graph structures has been verified in [2],[3] and elsewhere. The structure focusing on the LAYER structure has some aspects of novelty, but it seems to be a simple conventional combination idea and the degree of novelty is weak.
>
> We expand on why we believe our work is novel in the general response.
>
> > It is difficult to see what the similarity of the models is intended to achieve. The proposed method is similarity of the graph structure and may be similar for different models. The paper evaluates the similarity in application to pruning to demonstrate its validity. However, it is difficult to accept from the experimental results alone that the proposed method is a superior similarity simply because the implications that the similarity indicates are not clear. As a method for measuring the degree of change, such as model modifications, such as pruning, it is intuitively effective. However, as no uses have been indicated, the current situation is that there is only evidence for it as a pruning method. If the claim is that it is a superior pruning method, it should be clearly stated as such. In addition, if the pruning method is slaughtered and claimed, it should be compared with a generally sufficient advanced puruning method, for example [4],[5].
>
> We thank the referee for this feedback. As quoted in the general response, the aim of this paper is not to provide a competitive pruning method, rather to use the pruning application as a showcase for our zigzag framework. We agree with the referee that this only example might not be enough. We have run further experiments showing the potential of our framework, as described in the general response, and in Appendix D.
>
> > Clarify the claim point(s) for which the novelty of the contribution in this paper is sufficient
>
> We refer to the general response for this point.
>
> > Please clearly indicate in what sense the similarity of the proposal is valid, what use scenarios are covered (other than pruning) and whether it is sufficient evidence for this.
>
> We provide new experiments in this direction, showing that persistence similarity can be interpreted as indicating distinguishable phases in the model and discriminating over different types of input prompts (e.g. programming languages). See general response and Appendix D for further details.

---

> ### Comment · Reviewer_99nQ · 2024-11-27
>
> Thank you for your clarifications.
>
> As there is little time left in the discussion when the rebuttal was posted, there may be some misunderstanding due to written communication, and although it would be better to have several rounds of communication, I would like to make a final decision here this time. (I think there is a problem with the way the schedule is set and notified.)
>
> Regarding novelty, it is not clear from the rebuttal whether the novelty lies in modeling the evolution of general point clouds or in constructing a model for neural networks, but I think that the former is doubtful and the novelty of the latter is weak in simple application, so it is necessary to know what the solution to the problem of application was. Also, although it appears that you are arguing that there is novelty within the scope of TDA, that argument is not appropriate because the purpose is to learn about the characteristics of LLM. (In this conference, it is recognized that the improvement of TDA technology itself is out of scope. )
>
> Also, although it is stated in the appendix as being temporary, it is difficult to recommend it when you are not sure whether the final version will be satisfactory.
>
> Based on the above, I will maintain my score for now.

---

> > ### Comment · Area_Chair_JVP7 · 2024-11-27
> >
> > To provide additional context to the answer of the reviewer: While the paper itself is of course well within the scope of this conference, we can indeed not consider improvements to TDA-specific methods to be within scope. This is mostly due to the fact that we may simply not rely on a sufficiently large reviewer pool of experts that are capable of assessing to what extent the improvements are substantial.
> >
> > From my point of view, however, we have a rare case of having reviewers (including myself) who are all experts in computational topology! Thus, such an assessment would be feasible and I invite all reviewers to consider incorporating such information into their overall assessment.
> >
> > Thanks!
> >
> > — Your AC

---

> ### Comment · Reviewer_99nQ · 2024-11-27
>
> In response to AC's comment, I will add to what I said earlier.
>
> My intention is that, since the subject of this paper is the modeling of LLM characteristics, even if the method is new in relation to TDA, if a similar result can be achieved using a method that is not TDA, then the novelty of the method will be weakened. When I first read the paper, I thought it was claiming novelty in a broad range that was not limited to TDA, but in the rebuttal, it was phrased as being new as TDA, so I asked about it.

---

> > ### Comment · Area_Chair_JVP7 · 2024-11-27
> >
> > Thanks for the clarification! I also appreciate the willingness to engage with the authors and contextualise these issues.

---

> ### Author Response · Authors · 2024-11-27
> **Official Comment by Authors**
>
> We would like to thank the Area Chair and Reviewer 99nQ for replying to our comments even within such a short time. We also thank the Area Chair for clarifying things and considering a broader context. Here we hope to concisely clarify our viewpoint on what was raised above.
>
> - Our work's scope was **not** to provide novelty within the field of computational topology, which we agree i) it would be outside the scope of this conference, ii) we did not contribute with new theoretical results.
>
> - Indeed, our replies towards novelty within the field of TDA were a direct response to points raised by more than one reviewer, questioning the novelty of our approach within computational topology (e.g. zigzag formulation already existing). Our replies were meant to show that, within our specific application, there are a few elements of novelty that are worth considering.
>
> - Our work's scope is to provide a bridge between two fields (computational topology and interpretability of neural networks), and on the TDA side it should be viewed as a TDA **application**, while on the interpretability side it should be viewed as a new framework.
>
> - As a bridging paper, from our point of view it should have the following ingredients: i) an understandable connection between theoretical results (zigzag/TDA in this case) to the application field (interpretability of neural network), ii) a clear formulation of the framework and how is the application realized, iii) one or more practical experiments showcasing the effectiveness of the method.
>
> - As for point i), some reviewers recognized that we managed to present the zigzag formulation clearly.
>
> - Some of the discussion about TDA novelty has been about point ii): the novelties we argue about are a byproduct of the application we had to do using available TDA tools. Within this application, we introduced a few elements of novelty (knn filtration in zigzag, layers as time, persistence similarity) Combining these things, to our opinion, is a strength of our work.
>
> - A fair point that was raised in this respect was that there exist already several applications of TDA to interpret representations of neural networks, as cited in the relevant works section of our work. However, as argued in the general response, to the best of our knowledge, our work is the first to consider neural networks' representations as a whole dynamical system, while previous literature analyzed layers as single snapshots. Moreover, our work is the first to analyze internal representations of **transformer models** using TDA. We believe these are important points of novelty.
>
> - About point iii), we recognize that the initial submission might have been weaker than we intended. We believe we have strengthened this point with new experiments, as also recognized by reviewer n98n.
>
> We hope that these points clarify the positioning of this work in broader literature as a bridge between the fields of computational topology and interpretability of neural networks.

---

> ### Comment · Reviewer_99nQ · 2024-12-03
>
> Thank you for your sincere response to the discussion points. I find it interesting as an analytical method and discussion. The points “the first to consider neural networks' representations as a whole dynamical system” and “the first to analyze The points “the first to consider neural networks' representations as a whole dynamical system” and “the first to analyze internal representations of transformer models using TDA” are novelty as a means and should be combined with the traditional challenges for LLM analysis. It would be a good paper if it can be shown that it is experimentally effective, for example, by clearly showing what kind of problems (examples that cannot be identified) exist in not being able to capture the dynamical system as a whole. I will review the score, taking into account even the clarifications.

---

> > ### Author Response · Authors · 2024-12-03
> > **Official Comment by Authors**
> >
> > We appreciate reviewer 99nQ's constructive feedback, it surely helps improve our work.
> >
> > >The points “the first to consider neural networks' representations as a whole dynamical system” and “the first to analyze The points “the first to consider neural networks' representations as a whole dynamical system” and “the first to analyze internal representations of transformer models using TDA” are novelty as a means and should be combined with the traditional challenges for LLM analysis.
> >
> > This is a fair point: it is not enough to claim to have done a "new thing", it should also be useful. Our viewpoint here is that we showcased a few experiments that show that our framework can observe a few new aspects of the topology of internal representations that are linked to model behavior (e.g. sensitivity of topology to topics, sensitivity of performance to rate of change in topology), motivating further research using this framework.  We also note that we made an effort to present the framework in detail (along with well-documented code) and in simple terms, despite being zigzag persistence a non-trivial result (at least for non-computational topologists) which, in our opinion, has been underexploited so far.
> > Nevertheless, we understand the reviewer's viewpoint and appreciate their honesty.
> >
> > >It would be a good paper if it can be shown that it is experimentally effective, for example, by clearly showing what kind of problems (examples that cannot be identified) exist in not being able to capture the dynamical system as a whole. I will review the score, taking into account even the clarifications.
> >
> > While we understand the reviewer's general viewpoint that the paper should be experimentally effective (as remarked above), we disagree on the type of examples the reviewer is proposing we should have produced. Instead of showing what problem exists in not capturing the dynamical system as a whole, we showed what kind of advantages there are in capturing the dynamical system as a whole. Of course, some counterexample would be useful, but since the goal here is the interpretation of internal representations, we think there is not such a strong motivation to prove previous methods "wrong", or "incomplete", but more to introduce a new tool to observe new things, which is what we focused on. Nevertheless, we will take the reviewer's suggestion and consider adding a section on quantitative comparison with previous topological and geometrical tools for internal representation analysis.

---

### Official Review · Reviewer_rEUs · 2024-11-03

**Soundness:** 1
**Presentation:** 3
**Contribution:** 2
**Rating:** 3
**Confidence:** 4

**Summary:**

The paper uses zigzag persistence to empirically study LLMs.

**Strengths:**

Section 3 rigorously introduced the zigzag persistence.

**Weaknesses:**

The major problem of this paper is the lack of a problem. In fact, the word "problem" was not found in the whole paper at all.

Lines 520-521: "Our approach aims to provide a high-level geometrical and topological description of positional and relational changes across layers".

If this phrase in the conclusions should be considered a problem statement, then almost any empirical study using geometry and topology can fit this description.

Section 3 "Method" describes over 3 pages the known facts about zigzag persistence without even mentioning LLMs from the title of the paper.

The world "layer" appears in the paper 50+ times without a proper definition. Initially, this word is used in the context of neural networks but section 3 seems to assume that a layer is a cloud of unordered points in R^d, which is a standard input for computing persistence in TDA.

The "persistence similarity" introduced in (5) raises many concerns. First, the concept of "persistent similarity" appeared in the literature, e.g. https://www.intlpress.com/site/pub/pages/journals/items/cis/content/vols/0018/0004/a004/, but no past work on such similarity was cited.

More importantly, the denominator in (5) can vanish, which makes the persistence similarity S_p undefined. In this definition, it is still unclear if l_1,l_2 denote layers or point clouds because min/max of l_1,l_2 seem to be real numbers. If a layer is indeed a point cloud, the first metric axiom surely fails for l_1 and its translated image l_2, which have the same persistence.

Even if we consider point clouds under isometry, because persistence is an isometry invariant of a point cloud for standard filtrations of complexes, the first metric axiom also fails for infinitely many non-isometric point clouds, see J Applied Comp Topology 2024. Computational geometry has known much stronger isometry invariants of unordered point clouds for more than 20 years, see Boutin and Kemper, 2004.

Lines 402-403: "Note that the plot is not symmetric by definition (cfr. equation 5)".

Does it mean that the symmetry axiom fails?

The triangle axiom also seems unlikely to hold, so a proof is needed. If a distance fails the triangle axiom with any positive error, then results of clustering are not trustworthy as proved in https://ieeexplore.ieee.org/abstract/document/10574843?

The form on "soundness" asked to evaluate "the technical claims, experimental and research methodology and on whether the central claims of the paper are adequately supported with evidence".

No technical claims were found, no words "claim", "proposition", "theorem" in the paper. Here are the comments on informal claims about contributions.

Lines 74-76: "We propose a new metric to measure which topological features persist across the layers of an LLM."

If persistence similarity is called a metric, proofs of metric axioms are expected. The original persistence already measures "which topological features persist".

Lines 77-79: "By identifying layers with high persistence similarity, we prune redundant layers without significantly degrading performance".

The adjective "redundant" appears in the paper without explanation. If performance refers to percentages in Table 1, they degrade by 10%. Is it not significant?

Lines 80-82: "Our findings indicate that the behavior of persistent topological features and their similarities are consistent across different models, layers, and choices of hyperparameters of the framework".

To make this claim meaningful, the words "consistent" and "consistency" should be properly defined.

**Questions:**

Lines 523-524: "This approach allows for effective model pruning by identifying and removing redundant layers without significantly compromising performance"

All accuracies in Table 1 are between about 40% and 70%. Are these numbers enough to guarantee safety in real applications?

How are outputs of LLMs converted into point clouds in the experiments? What is the meaning of persistence in terms of word tokens or other language-related concepts?

Should the review mention the previous work explaining LLMs as stochastic parrots at https://dl.acm.org/doi/abs/10.1145/3442188.3445922 and in stronger terms at https://link.springer.com/article/10.1007/s10676-024-09775-5?

---

> ### Author Response · Authors · 2024-11-26
> **Reply to Reviewer rEUs**
>
> ## Reply to rEUs
>
> > The major problem of this paper is the lack of a problem. In fact, the word "problem" was not found in the whole paper at all.
> Lines 520-521: "Our approach aims to provide a high-level geometrical and topological description of positional and relational changes across layers".
> If this phrase in the conclusions should be considered a problem statement, then almost any empirical study using geometry and topology can fit this description
>
> We thank the referee for the feedback on the contextualization of our manuscript. We have now clarified more clearly what problem we are trying to address both in the introduction and in the conclusions.
>
> >Section 3 "Method" describes over 3 pages the known facts about zigzag persistence without even mentioning LLMs from the title of the paper.
>
> We thank the referee for pointing out that we should be careful in contextualizing where our methodology is applied. We kindly point out that an explicit reference is made in the introductory paragraph of the method section. After defining the connection to LLMs for defining our point clouds, we believe our methodology explanation should be general. We refer to our general response for the novelty aspects of our zigzag algorithm.
>
> >The world "layer" appears in the paper 50+ times without a proper definition. Initially, this word is used in the context of neural networks but section 3 seems to assume that a layer is a cloud of unordered points in R^d, which is a standard input for computing persistence in TDA.
>
> We thank the referee for feedback on clarity of presentation. In the same introductory part of section 3, we have expanded on what we mean by layer.
>
> > The "persistence similarity" introduced in (5) raises many concerns. First, the concept of "persistent similarity" appeared in the literature, e.g. https://www.intlpress.com/site/pub/pages/journals/items/cis/content/vols/0018/0004/a004/, but no past work on such similarity was cited.
>
> We thank the referee for pointing out previous literature. Indeed, past work have introduced the terminology “persistence similarity” in different contexts, and for different uses. We added a footnote citing the work to clarify possible misunderstandings in the text.
>
> > More importantly, the denominator in (5) can vanish, which makes the persistence similarity S_p undefined.
>
> As the referee correctly points out, equation 5 is ill-defined for zero betti numbers at a given layer and given dimension. By definition, persistent similarity should be 0 whenever betti numbers at either l1 or l2 are 0. We added a footnote explaining this case, but did not modify the equation for readability.
>
> >  In this definition, it is still unclear if l_1,l_2 denote layers or point clouds because min/max of l_1,l_2 seem to be real numbers.If a layer is indeed a point cloud, the first metric axiom surely fails for l_1 and its translated image l_2, which have the same persistence.
> Even if we consider point clouds under isometry, because persistence is an isometry invariant of a point cloud for standard filtrations of complexes, the first metric axiom also fails for infinitely many non-isometric point clouds, see J Applied Comp Topology 2024. Computational geometry has known much stronger isometry invariants of unordered point clouds for more than 20 years, see Boutin and Kemper, 2004.
>
> l1 and l2 here indicate layer indices, thus they are integer numbers. It would seem consistent with the rest of the notation used in previous definitions, but we would appreciate any feedback for a better notation. We do not understand a referrence to isometry in this context, the first reference proposed by the referee seems incomplete. As for the second reference, we apologize for not understanding the specific relevance to our context.
>
> > Lines 402-403: "Note that the plot is not symmetric by definition (cfr. equation 5)".
> Does it mean that the symmetry axiom fails?
> The triangle axiom also seems unlikely to hold, so a proof is needed. If a distance fails the triangle axiom with any positive error, then results of clustering are not trustworthy as proved in https://ieeexplore.ieee.org/abstract/document/10574843?
>
> In this context, we refer to an asymmetry over the arguments of the function (l_1,l_2), which we remind are indices to layers. Thus, this would mean that swapping l_1, l_2 provides different values of the functions.

---

> > ### Author Response · Authors · 2024-11-26
> > **(continuation)**
> >
> > > The form on "soundness" asked to evaluate "the technical claims, experimental and research methodology and on whether the central claims of the paper are adequately supported with evidence".
> > No technical claims were found, no words "claim", "proposition", "theorem" in the paper. Here are the comments on informal claims about contributions.
> >
> > > Lines 74-76: "We propose a new metric to measure which topological features persist across the layers of an LLM."
> > If persistence similarity is called a metric, proofs of metric axioms are expected. The original persistence already measures "which topological features persist".
> >
> > We thank the referee and apologize for the improper use for the word "metric". We have rephrased the occurrences to solve the ambiguity.
> >
> > > Lines 77-79: "By identifying layers with high persistence similarity, we prune redundant layers without significantly degrading performance".
> > The adjective "redundant" appears in the paper without explanation.
> >
> > We agree with the referee that the use of "redundant" was previously undefined. Given that the word was not needed for the clarity of the sentences where it was used, we simply removed it.
> >
> > >If performance refers to percentages in Table 1, they degrade by 10%. Is it not significant?
> >
> > We thank the referee for raising a fair point. As explained in the general response, the aim of the pruning layer algorithm using zigzag was to showcase the methodology, rather than providing a competitive method in this context. Nevertheless, we note that other experiments with other methods have shown similar degradations and are commonly accepted to considered as small degradations in this specific set of tests.
> >
> > >Lines 80-82: "Our findings indicate that the behavior of persistent topological features and their similarities are consistent across different models, layers, and choices of hyperparameters of the framework".
> > To make this claim meaningful, the words "consistent" and "consistency" should be properly defined.
> >
> > We thank the referee for pointing out an undefined expression. We have replaced all occurrences were it created ambiguity.
> >
> > >Lines 523-524: "This approach allows for effective model pruning by identifying and removing redundant layers without significantly compromising performance"
> > All accuracies in Table 1 are between about 40% and 70%. Are these numbers enough to guarantee safety in real applications?
> >
> > We thank the referee for raising a fair point. As this is an experiment to showcase an application of our method, for safe applications of this pruning we would need to employ further refinements, like fine-tuning. We deserve this to future work.
> >
> > >How are outputs of LLMs converted into point clouds in the experiments? What is the meaning of persistence in terms of word tokens or other language-related concepts?
> >
> > We currently explain how LLMs are converted to point clouds in introduction to section 3. As word tokens are embedded as vectors in a D-dimensional space, persistence represents the survival of a given $p$-cycle formed by several word-tokens as points across layers.
> >
> > >Should the review mention the previous work explaining LLMs as stochastic parrots at https://dl.acm.org/doi/abs/10.1145/3442188.3445922 and in stronger terms at https://link.springer.com/article/10.1007/s10676-024-09775-5?
> >
> > We do not believe these references are relevant to the scope of our paper, and including them would necessarily imply including several other papers that do not fit in the overall context of the paper.

---

> > > ### Comment · Reviewer_rEUs · 2024-11-28
> > > **Thank you for the reply**
> > >
> > > >"it has not yet been recognized that the internal representations of LLMs can essentially be viewed as dynamic point clouds evolving in time (layers). As pre-trained LLMs process inputs, they transform these point clouds within the representation space layer by layer, capturing essential features and relationships throughout the model’s depth. Thus, it is natural to interpret these transformations as an evolving discrete dynamical system. To address this problem,"
> > >
> > > The revision contains the word "problem" once in the quoted paragraph above but these phrases do not formalise a problem because anyone can say: "Ok, I interpret these transformations as an evolving discrete dynamical system". Is the problem solved?
> > >
> > > To formulate a problem, one could first define all necessary concepts and then state conditions that should be satisfied by a solution. Otherwise, any algorithmic output can be called a solution. So the main concern remains the same: "The major problem of this paper is the lack of a problem."
> > >
> > > >swapping l_1, l_2 provides different values of the functions.
> > >
> > > Then the persistence similarity fails the symmetry axiom of a distance metric. Can you propose a metric satisfying all axioms?

---

> > > > ### Author Response · Authors · 2024-11-29
> > > > **Reply to Reviewer rEUs**
> > > >
> > > > >The revision contains the word "problem" once in the quoted paragraph above but these phrases do not formalise a problem because anyone can say: "Ok, I interpret these transformations as an evolving discrete dynamical system". Is the problem solved?
> > > > To formulate a problem, one could first define all necessary concepts and then state conditions that should be satisfied by a solution. Otherwise, any algorithmic output can be called a solution. So the main concern remains the same: "The major problem of this paper is the lack of a problem."
> > > >
> > > > We thank the reviewer for providing further feedback on the correct formulation of the problem. Since we can't modify the draft at this point, we reply directly here intending to modify the draft accordingly whenever possible.
> > > >
> > > > The problem we would like to address can be decomposed into several layers of generality. The general problem we are addressing is that LLMs (and neural networks in general) are not interpretable: we do not understand what goes on inside, we can't evaluate incorrect or unsafe behavior and we cannot systematically optimize efficiency. Within efforts to address this problem, a series of works have focused on analyzing internal representations using various strategies. Within these strategies, some have proposed to analyze the geometry and topology of the manifold on which representations live. Our work is positioned within this last step. The first few paragraphs of the paper were meant to introduce this, though we realize we might not have been clear enough.
> > > >
> > > > Now, within these efforts to study the geometry and topology of internal representations, we realized there was a gap in the literature: previous work has considered studying internal representations at each layer independently from each other, and then combined the information gathered in each layer to provide a coherent insight on internal representation. Our framework, instead, considers the set of layers as a whole. We believe this to be an important step further in this effort to study the geometry and topology of internal representations for the interpretability of LLMs.
> > > >
> > > > We believe these sentences define the problem we would like to solve (in several layers of generality) and the solution we are proposing. The paper is a presentation on how the framework should be formulated and provides a few examples to show that it indeed provides new interpretable aspects of LLMs. Of course, we do not solve the problem of LLMs interpretability altogether. Moreover, we are not done with demonstrating that our proposal is effectively a (or part of the) solution of the problem. Nevertheless, we believe the progress we make enables further studies in this new direction, as recognized by another reviewer.
> > > >
> > > > As a side note, we understand, since the paper utilizes several concepts in computational topology, that the reviewer was expecting a more rigorous and mathematical definition of the statements we make, and the problem itself. However, as explained in the general response, this is not a paper in computational topology. The reviewer might correctly argue that a rigorous definition of the problem should not be given only for papers that are strictly within mathematical subjects. We generally agree with this argument, though for this paper we decided to focus on an intuitive presentation of the problem, as a strategy to make the paper accessible for a wider audience.
> > > >
> > > > > Then the persistence similarity fails the symmetry axiom of a distance metric. Can you propose a metric satisfying all axioms?
> > > >
> > > > If defined as a distance metric, persistence similarity would indeed be ill-defined. However, as specified in the general and specialized response, we have changed all occurrences where we improperly used the term "metric" substituting with "topological descriptor". We believe the misunderstanding resided in the different uses of the wording "metric" in different contexts. While we agree with the reviewer that the primary use should be in the context of a mathematical definition, it is also used in interpretability literature close to the ones covered by our paper, ( it was also used within these reviews by two reviewers.). We hope this clarification solves the misunderstanding.

---

### Official Review · Reviewer_n98n · 2024-11-03

**Soundness:** 3
**Presentation:** 3
**Contribution:** 2
**Rating:** 6
**Confidence:** 2

**Summary:**

This paper uses zigzag persistence, an approach from TDA, to analyze the hidden representations of LLMs. With these tools, they track the evolution of topological properties of LLM hidden representations across layers. They make numerical observations such as increased persistence of topological features formed in later layers, and high similarity between representations in later layers. As an application, they prune layers based on persistence similarity.

**Strengths:**

1. Generally good and clear explanation of the requisite background and method.
2. Experiments show some insights into several open-weight language models of interest, across datasets that people care about (both for computing the metrics, and for testing downstream accuracy after pruning).

**Weaknesses:**

1. Besides the layer pruning application, the qualitative and quantitative insights from these topological features are not particularly interpretable or grounded. We can measure how the number of $k$-cycles evolves, but there are no interpretable quantities that come out of this, besides similarities between layers. To alleviate this, perhaps further discussion of motivation and related work (why should people use TDA tools to analyze internal representations), or interpretability work (what kind of 1-cycles form, and on what kind of data) could be interesting.
2. Utility in one downstream application (layer pruning) is questionable (the simpler methods from prior work do solidly and are not consistently outperformed).
3. The use of zigzag persistence in layer pruning only depends on the similarity matrix formed by the method. Other simple baselines could be considered for making similarity matrices.

**Questions:**

1. Have you looked at how topological features vary across different data samples / domains? For instance, are there any interesting data features that form between representations of tokens from certain programming languages?
2. Could you provide a bit more information on where the embeddings are extracted from (this is currently described in Section 4.1 as "each prompt is processed so that the last token is extracted at each normalization layer and the final normalization applied to the output layer.") Do you take the embedding before or after the normalization layer? Also, does this mean you take two embeddings from each block (one from the attention normalization, one from the MLP normalization)? Also, some discussion on how this compares to how other works extract LLM hidden representations  would be nice.

---

> ### Author Response · Authors · 2024-11-26
> **Response to Reviewer n98n**
>
> ## Reply to N98N
>
> > Besides the layer pruning application, the qualitative and quantitative insights from these topological features are not particularly interpretable or grounded. We can measure how the number of k-cycles evolves, but there are no interpretable quantities that come out of this, besides similarities between layers. To alleviate this, perhaps further discussion of motivation and related work (why should people use TDA tools to analyze internal representations), or interpretability work (what kind of 1-cycles form, and on what kind of data) could be interesting.
>
> We thank the referee for the suggestion. We believe we improved the interpretability of our work, as explained in the general response, by performing a series of experiments which help in connecting our topological descriptors with model behavior.
>
> As for interpreting what kind of 1-cycles form, and on what kind of data, we would like to point out that the current state of research in zigzag persistence does not allow to unambigously track single topological features. Working towards this direction would certainly be interesting but outside the scope of this work.
>
> > Utility in one downstream application (layer pruning) is questionable (the simpler methods from prior work do solidly and are not consistently outperformed).
>
> We agree with the referee that the layer pruning might be a questionable downstream application and that our methodology does not always outperform existing methods.
> In fact, the novelty of this paper is to show a new approach on the study of the internal workings of LLMs, in particular the application to the analysis of internal representations and the evolution of the manifold they lie in.
>
> Nevertheless, we have performed further experiments to make the utility of our approach more robust, see general response and appendix D.
>
> > The use of zigzag persistence in layer pruning only depends on the similarity matrix formed by the method. Other simple baselines could be considered for making similarity matrices.
>
> The referee raises a fair point. Our similarity criterion contains strictly more information than usual similarity matrices because it depends explicitly on the trajectory of features, while other similarity matrices are only computed through pairwise comparison. Our new experiments provide evidence of some of the information contained in our similarity measure (see Figures 8,9,10).
>
> > Have you looked at how topological features vary across different data samples / domains? For instance, are there any interesting data features that form between representations of tokens from certain programming languages?
>
> We thank the referee for suggesting an interesting experiment. We take the suggestion by doing the average similarity of three distinct datasets where the language difference is embedded in the semantic of the prompts, in this case a mathematical dataset, a code dataset from github and SST which contains movie reviews (and was already present in the work). We can explicitly see that datasets using technical language (e.g. special characters) show two distinct increasing similarity stages, while general text only show one increase phase. We go into further details in the general response, and Appendix D.
>
> > Could you provide a bit more information on where the embeddings are extracted from (this is currently described in Section 4.1 as "each prompt is processed so that the last token is extracted at each normalization layer and the final normalization applied to the output layer.") Do you take the embedding before or after the normalization layer? Also, does this mean you take two embeddings from each block (one from the attention normalization, one from the MLP normalization)? Also, some discussion on how this compares to how other works extract LLM hidden representations would be nice.
>
> We thank the author for pointing this out, we take the embeddings one time after the normalization layer. In particular we followed the procedure from [4].

---

> > ### Comment · Reviewer_n98n · 2024-11-26
> >
> > Thanks for the very reasonable response, and for the clarifications. These additional, more fine-grained experiments are nice and interesting. I like the comparison across different datasets. I'll raise my score from a 5 to a 6.

---

### Official Review · Reviewer_bMgZ · 2024-11-04

**Soundness:** 2
**Presentation:** 3
**Contribution:** 2
**Rating:** 3
**Confidence:** 3

**Summary:**

This paper proposes a novel topological data analysis method based on zigzag persistence, which describes data undergoing dynamic transformations across layers within the large language models. The proposed persistence similarity is used to quantify the persistence and transformation of topological features throughout the model layers and provide insights to prune redundant architecture modules without significantly degrading performance. Experimental results indicate that the behavior of persistent topological features and their similarities are consistent across different models, layers, and choices of hyperparameters of the framework, suggesting a certain degree of universality in the topological structure of LLM representations.

**Strengths:**

- The paper is clearly written, making complex concepts accessible to the reader.
- The application of zigzag filtration to the internal representation of LLM is novel and insightful.
- The approach to zigzag persistence is firmly rooted in theoretical foundations, providing a robust framework that enhances its applicability and relevance in the field.

**Weaknesses:**

- The zigzag filtration, and persistence images are previously well-defined and existing. This raises concerns about the originality and depth of the theoretical and technical contributions presented in the paper.
- The performance gains over existing pruning methods, as shown in Table 1, appear marginal. Including a significance test would enhance the robustness of these findings.
- While the paper provides a geometric interpretation of the topological features, it is not clear how these features can be directly interpreted in the context of language models. The paper could benefit from a more in-depth discussion on the interpretability and implications of the observed topological properties.
- The zigzag algorithm is computationally expensive, especially for large datasets and high-dimensional representations. The paper should discuss strategies to optimize the algorithm's performance and make it more scalable.

**Questions:**

- Figure 5 is somewhat unclear. Are the 10% (orange) segments included within the 20% (yellow) segments in most cases? The authors might consider a more effective visualization method to enhance clarity.
- Since other similarity measures can also characterize layer-wise behavior, what specific advantages does the proposed persistence similarity offer over these existing methods?
- What is the computational complexity of the proposed persistence similarity in comparison to other existing methods?
- Can the authors provide insights or a high-level intuition regarding the consistent patterns observed in Figure 4?

---

> ### Author Response · Authors · 2024-11-26
> **Reply to Reviewer bMgZ**
>
> ## Reply to bMgZ
>
> > The zigzag filtration, and persistence images are previously well-defined and existing. This raises concerns about the originality and depth of the theoretical and technical contributions presented in the paper.
>
> While we do agree with the referee that zig zag filtrations and persistence images have been used previously in the literature, we believe our paper introduces a few aspects which, combined, do represent an element of novelty with respect to previous research. We have summarized these points in the general response.
>
> > The performance gains over existing pruning methods, as shown in Table 1, appear marginal. Including a significance test would enhance the robustness of these findings.
>
> We thank the reviewer for suggesting to enhance the robustness of our findings with a significance test. Considering the computational effort required to run a statistically solid significance test, we did not have time to provide trustable results within the resubmission window.
> Nevertheless, we performed a rough test with the data we already have: we consider each performance evaluation as a stochastic variable, being each benchmark and model a different observation of the system at fixed percentage cut (10% or 20%). We compute the mean and standard deviation of these observations, and observe that our method is not significantly different from the other two methods we refer to in the paper. We believe that without a proper statistical treatment, this result should not be included as a proof that our methodology performs as well as other methods. We are open to include a more solid test if required (and given sufficient time).
>
> > While the paper provides a geometric interpretation of the topological features, it is not clear how these features can be directly interpreted in the context of language models. The paper could benefit from a more in-depth discussion on the interpretability and implications of the observed topological properties.
>
> We thank the referee for suggesting an improvement to our manuscript. We have run a series of more detailed experiments to better understand what our topological descriptors tell about model behavior. We describe these tests in the general response and in resubmitted version. See Appendix D.
>
> >The zigzag algorithm is computationally expensive, especially for large datasets and high-dimensional representations. The paper should discuss strategies to optimize the algorithm's performance and make it more scalable
>
> We thank the referee for pointing out we should make clearer what is the computational complexity of ZigZag. Currently, the Fast ZigZag algorithm [3] allows to explore high-dimensional and large datasets. The computational cost is explained Theorem. 22 of [3].
> Moreover, to avoid the development of the filtration with Vietoris-Rips that would require the expensive and not trivial task of findind the right radius over multiple point clouds, we use a $k_{\rm NN}$-based filtration, as discussed in the paper. Note that in the appendix B we discuss a different way to build the filtration with $k_{\rm NN}$ and Vietoris-Rips.
>
> > Figure 5 is somewhat unclear. Are the 10% (orange) segments included within the 20% (yellow) segments in most cases? The authors might consider a more effective visualization method to enhance clarity.
>
> We thank the referee for the suggestion, we improved the visualization of Figure 5.
>
> > Since other similarity measures can also characterize layer-wise behavior, what specific advantages does the proposed persistence similarity offer over these existing methods?
>
> The specific advantage of our method is that it follows the full trajectory of features across layers. This is different from previous similarity measures, which were only considering pairwise comparisons. We believe that this advantage emerges from the new experiments we have run, see Appendix D.
>
> > What is the computational complexity of the proposed persistence similarity in comparison to other existing methods?
>
> The persistence similarity is mainly composed by the cost of the Fast ZigZag algorithm that is quasi-linear. For Fast ZigZag computation refer to [3].
>
> > Can the authors provide insights or a high-level intuition regarding the consistent patterns observed in Figure 4?
>
>
> We thank the referee for this question, the answer is provided in the general response and it has been added to the main text and in Appendix D.

---

### Author Response · Authors · 2024-11-26
**General Response to all reviewers**

# General response
We thank the reviewers for carefully reading our draft, recognizing its strengths and providing feedback on its
weaknesses. These suggestions have helped us to greatly improve our work. Here we provide a general response to
the main points raised and reply more in detail in each thread.

**Strengths**.

The reviewers recognized the novelty and potential impact of applying topological data analysis
(TDA), specifically zigzag persistence, to the study of internal representations in large language models (LLMs).
We appreciate the acknowledgment of our work for bringing a fresh perspective to model analysis and geometric
interpretation through robust theoretical foundations. A strong point raised by more than one review has been the
clarity and presentation of our paper, making complex concepts accessible.

**Weaknesses**

Some of the reviewers consistently highlighted a few key areas for improvement:

- Novelty of zigzag approach, being already developed and employed in other applications.
- The work does not concretely link the topological features and observables to interpretable insights within
the LLM context.
- Layer pruning using persistent similarity does not seem superior to other methods, and its practical utility
has not been investigated.
- Overall, further experiments or applications of the framework would have strengthened the work.

In summary, our work is criticized for not developing/being novel enough either of the two subfields it's talking about, i.e. TDA and layer pruning. This is a fair concern, so we address it in the following response and make modifications to the draft accordingly. To preserve the integrity of the original draft, we have momentarily included results of new experiments in the appendix and not in the main body.

## Why ZigZag on LLMs?
On a high level, using topology to study how representations change in neural networks has been recognized
for a while (e.g. [1]) as a promising approach. As point clouds embedded in a
representation space, most of the existing work has considered TDA as a promising avenue for making progress
in this direction. In the context of internal representations of neural network, there is a sequence (layers) of these point clouds, and previous works analysing these sequences using TDA have tried to gather insights on how these point clouds change with respect to to different models, different tasks, and within the sequence. However, to the best of our knowledge, none of the previous works have built a TDA framework
that models the sequence of point clouds as a unique evolving system, rather than analyzing each step of the sequence separately and comparing results. This means that we can follow topological features across layers tracking the trajectory of these features through each layer. This implies that, for instance, swapping two layers is not a symmetry of our framework, i.e. our descriptors change, differently than previous work.

Overall, while not rigorously defining a new theoretical TDA framework, our paper serves as a crucial conceptual brige to open a range of possibilities for an interpretable set of experiments to be done on internal representations.

Moreover, on a practical and more technical level, our zig-zag algorithm introduces a few novel points which we believe are important in this context:

- **Choice of filtration**: we introduce a $k_{\rm NN}$ graph-based filtration, which is relatively novel in previous TDA literature [1], and never used in the context of ZigZag. This choice is motivated by the fact that the point clouds are largely transformed from one layer to another (e.g. space rotations, translations, stretching of distances). For this reason, conventional ways to set up a filtration (e.g. Vietoris Rips complexes) are not viable. In addition, we provide a discussion in the appendix on how to combine the $k_{\rm NN}$ graph with a Vietoris Rips complex, which is also novel in the context of zigzag filtrations.

- **Layers as Time snapshots**: while the theoretical framework for zigzag has been developed in depth, its true potential seems still underexploited, as applications of ZigZag are still relatively few. As cited in the paper and one of the reviewers, current applications are in very different contexts/uses. We believe the interpretation of transformers as a dynamical system which evolves points clouds in "time" as layers is novel in the context of TDA.

- **Persistence Similarity**: persistence similarity is a new topological descriptor which we introduce for the first time. The layer pruning is just a showcase of its use, while we expect other potential applications. We show a couple of explorations in the new paper appendix D (and explained below).

---

> ### Author Response · Authors · 2024-11-26
> **(continuation)**
>
> ## Additional experiments for interpretability
>
> Given what introduced before, the application of our framework to a layer pruning task was meant to showcase the method, rather than providing a competitive pruning strategy by itself. We recognize we could have been more clear in this framing, and also that overall we should have been more extended in the description/range of the showcased experiment. Towards improving this part, we performed a series of experiments:
>
> ### In-depth analysis of average similarity
>
> Following the suggestions proposed by the reviewers, we extend our analysis of what our topological descriptor, average similarity, shows. Looking at Figure 4 (right) of our manuscript, we see that average similarity for LLMs is characterized by three phases:
>
> - **An increasing phase**, lasting until middle layers. The rate of increase is constant in this range, and across models, i.e. it does not depend on the nature of the model, its size and very weakly on the dataset used (cfr. Figure 7 (right) in the Appendix). It does depend on the underlying filtration (cfr. Figure 4 (left)). We have verified that for different values of $k_{\rm NN}$, the universality of the increase across models is conserved. The positive rate of increase means that average persistence of cycles increases, i.e. the transformer architectures are progressively learning features about the dataset that they want to retain.
>
> - **A plateau phase**, during which the rate of retention saturates. During this phase, we verified that layers with a high similarity are less important for performance.
>
> - **A decreasing phase** during the last few layers of the models. During this phase, the rate of retention changes again with an opposite trend as in the early layers: features are progressively destroyed.
>
> These observations motivate a few experiements:
>
> #### Experiments
>
> - To test the increasing phase, we compute the average similarity for a subset of data which share a common topic. The idea is to test whether this phase can change in extension or rate. As suggested by one of the reviewers, we consider datasets of mathematical problems, a general dataset of code (Figure 8) and 5 datasets of code of just one programming language each (Figure 9). We describe in detail this experiment in Appendix D.1. In short, we find that for these datasets, the increasing phase is split in two, with a previously unseen plateau in the middle. We argue that this behavior is triggered by special characters generally not used in conventional human language. We confirm this expectation showing that for programming languages that are less verbose (e.g. markdown) the splitting does not take place.
>
> - A test of the plateau is performed by shuffling tokens within prompts of the datasets, as a way of destroying the structure and semantic coherence of the prompt, without modifying its unigram frequency distribution. In Figure 8 we show how the plateau is modified by this change across two different datasets: the increase phase is shorter and the plateau is much lower in similarity.
>
> - An overall test of these phases is performed by pruning blocks of 5 adjacent layers with a sliding window of 2 and testing the model on
> a benchmark (MMLU). The scope of this experiment is to show how performance is directly affected by these phases. We show details of this experiments in Appendix and the result in Figure 10 (left panel). We see that performance is random choice during the increasing phase and it pleateaus close to the maximum during the plateau phase. As an interesting finding we see a drop in performance right in correspondance of the decreasing phase, which we investigate further by changing the block and sliding sizes, see Figure 10 (middle and right panels). This finding deserves a closer investigation, which we leave for future work.
>
> ## Errata Corrige
> We fixed the plot of Figure 2 which contained a typo of $k_{\rm NN}$ equal to 10 instead of 15.
> ## References
> [1] Ming Quang Le and Dane Taylor,Persistent homology with k-nearest-neighbor filtrations reveals topological convergence of PageRank. Foundations of Data Science.
>
> [2] C. Olah, “Neural networks, manifolds, and topology.”.
>
> [3] Fast Computation of ZigZag Persistence. Tamal K.Dey, Tao Hou. 30th Annual European Symposium on Algorithms (ESA 2022) https://doi.org/10.4230/LIPIcs.ESA.2022.43
>
> [4] The Geometry of hidden representation of large transformer models, L. Valeriani, D.Doimo , F. Cuturello, A. Laio, A. Ansuini, A. Cazzaniga. NeurIPS, 2023

---

### Meta-Review · Area_Chair_JVP7 · 2024-12-17

**Metareview:**

This submission presents a new method for tracking the evolution of geometrical-topological features through the layers of a large language model. Specifically, using zigzag persistent homology (a method from the nascent field of computational topology), the manuscript develops *persistence similarity*, a score for assessing common topological features (here, to be understood as features like connected components, cycles, ...; the method is capable of calculating such features across multiple spatial scales) between layers. Based on this score, the manuscript outlines a new algorithm for the identification (and subsequent pruning) of (partially) redundant layers, exhibiting promising results in comparison to other methods.

Coming from a background of computational topology myself, I appreciate the two primary conceptual *strengths* of the paper, to wit:

1. The paper is well-written and aims to explain its methods, thus exhibiting a high degree of clarity
2. The use of TDA or TDA-like methods in this context is highly original, and the topic is highly relevant to the community as a whole

At the same time, I also have some concerns and observed several *weaknesses* of the work, namely:

1. A lack of experiments that are compelling to the larger ML community. I realise that this sounds very negative and I hope I can take away some of the sting of these words. My point is that this paper, which makes creative use of TDA techniques to address a compelling problem is interesting _per se_ for people with a background in TDA. However, the goal of ICLR (and of ML conferences in general) is to have works that are compelling by providing new insights for a larger community and ideally considerably advance the state of the art. The authors are honest about the shortcomings of their method—which I appreciate—and mention that they cannot show statistical significance of the results.
2. The derivation of interpretable/explainable results from the method, along with a better justification for the use of topological methods. Again, I realise that this sounds overly negative, so I want it to be clear that I share the enthusiasm about TDA techniques and zigzag persistence in particular, but I need to consider the broader ramifications of using the "topology hammer" in front of a broader ML audience. Given the computational but also the conceptual complexity of the method, it is unfortunately our "onus" to also ensure that we justify the choice of method. Here, potential justifications could be interpretability/explainability or robustness but none of these aspects is considered so far.

It is for these reasons that I have to suggest rejecting the paper, and I hope the authors believe me when I say that this decision was not reached ligthtly. On the contrary: I am biased towards having _more_ "TDA goodness" at ML conferences, but I believe that in its current form, this work is not quite ready yet. Moving forward, I would suggest that the authors invest in expanding the experimental section, both in depth and in breadth. With a few modifications (which, unfortunately, will amount to some additional experiments), this has the potential to be a strong, impactful contribution.

I also want to emphatically state that such work _is_ in-scope with ICLR and related conferences, as well as other A*-conferences with a focus on NLP, and I sincerely hope that the authors can benefit from the discussion phase.

**Additional Comments On Reviewer Discussion:**

Reviewers appreciated the clarity and accessibility of the work, with minor suggestions for improvements concerning those readers that are not yet familiar with TDA (`n98n`, `rEUs`, `TVBg`); they also lauded the originality of zigzag persistent homology for this application (`bMgZ`, `99nQ`, `TVBg`). Most concerns were raised about the experimental setup and the (conceptual) complexity of method.

*Summary of the discussion:*

- Reviewer `bMgZ` raised concerns about the strength of the claimed gains; this could be partially alleviated—the authors also provided extensive text to better contextualise the method.
- Reviewer `n98n`, who feels positively about this paper, raised questions that were all addressed during the rebuttal, except for a more abstract question concerning the utility of the method—I alluded to this issue above as well.
- Reviewer `rEUs` feels negatively about this paper and raises concerns about the underlying methodology, i.e. persistent homology, _per se_, as well as some issues with the clarity for an ML audience. I do not endorse all these views; in particular I believe that the authors contextualised their work nicely and made it sufficiently accessible to a machine-learning audience. Several of the points can be explained by differences in terminology between different fields, and have since been rectified by the authors.
- Reviewer `99nQ` requested an additional contextualisation of the method as well as a comparison to related literature. These have been addressed by the authors, but the ensuing discussion resulted in additional concerns about the experimental efficacy, which could _not_ all be addressed yet—mostly because of the time required to perform additional experiments.
- Reviewer `TVBg` raised some questions resulting from a misunderstanding, which could be alleviated by the authors, as well as a concern about the sensitivity/overall efficacy of the method, which was not fully addressed during time concerns.

My final decision is primarily built on the concerns about the efficacy/utility of the method, and I am confident that these concerns can rectified by the authors. At the same time, given the state of the experimental section, I strongly believe that another round of reviews will be required for this paper. As outlined above, I hope the authors find the discussion useful in improving their paper; I certainly see several salient points raised by reviewers.

---

### Decision · Program_Chairs · 2025-01-22

Reject